# Scaling Gaussian Process Regression with Full Derivative Observations

**Daniel Huang**                                                                    *dan@base26labs.com*
*Base26, CA, USA*

**Reviewed on OpenReview:** *https://openreview.net/forum?id=fbonXp38r9*

## Abstract

We present a scalable Gaussian Process (GP) method called DSoftKI that can fit and predict full derivative observations. It extends SoftKI, a method that approximates a kernel via softmax interpolation, to the setting with derivatives. DSoftKI enhances SoftKI's interpolation scheme by replacing its global temperature vector with local temperature vectors associated with each interpolation point. This modification allows the model to encode local directional sensitivity, enabling the construction of a scalable approximate kernel, including its first and second-order derivatives, through interpolation. Moreover, the interpolation scheme eliminates the need for kernel derivatives, facilitating extensions such as Deep Kernel Learning (DKL). We evaluate DSoftKI on synthetic benchmarks, a toy n-body physics simulation, standard regression datasets with synthetic gradients, and high-dimensional molecular force field prediction (100-1000 dimensions). Our results demonstrate that DSoftKI is accurate and scales to larger datasets with full derivative observations than previously possible.

## 1 Introduction

A convenient feature of using a Gaussian Process (GP) to approximate functions is its ability to incorporate derivative observations since the derivative of a GP is also a GP. This enables more informative modeling when derivative data is available (*e.g.*, in the physical sciences). However, utilizing a GP with derivative observations, abbreviated GPwD, for regression faces significant scalability challenges. In particular, vanilla GPwD inference has time complexity $\mathcal{O}(n^3 d^3)$ where $n$ is the number of data points in a dataset and $d$ is the dimensionality of the data. Thus, GPwDs have scaling challenges in both $n$ and $d$.

To alleviate these challenges, scalable GP regression methods such as a *Stochastic Variational GP* (SVGP) (Hensman et al., 2013) have been extended to the setting with derivatives (DSVGP) (Padidar et al., 2021). It uses $m \ll n$ *inducing points* (Quinonero-Candela & Rasmussen, 2005; Snelson & Ghahramani, 2005) and achieves a time complexity of $\mathcal{O}(m^3 d^3)$ for posterior inference. Since the cubic scaling in $d$ can be prohibitive, a *DSVGP with directional derivatives* (DDSVGP) (Padidar et al., 2021) introduces $p \ll d$ *inducing directions*, an analogue of inducing points for dimensions, to achieve a time complexity of $\mathcal{O}(m^3 p^3)$ for posterior inference.[1] However, this comes at the cost of directly predicting derivatives and introducing further approximations.

In this paper, we present a GPwD method called DSoftKI[2] that can fit and predict all derivative information (Section 4). It is an extension of SoftKI (Camaño & Huang, 2025), a scalable GP method that approximates a kernel via softmax interpolation from $m \ll n$ interpolation points whose locations are learned. Thus, it blends aspects of kernel interpolation (Wilson & Nickisch, 2015) which introduces kernel interpolation from a structured lattice with variational inducing point methods (Titsias, 2009) where inducing points are adapted to the dataset. To handle gradient information, DSoftKI modifies SoftKI's interpolation scheme by replacing its global temperature vector with local temperature vectors associated with each interpolation

---

[1]This assumes $mp^2 > d$ which no longer holds around $d = 2000$ for $m = 500$ and $p = 2$. See Appendix E for details.
[2]Code available at `https://github.com/base26labs/dsoftki_gp`.

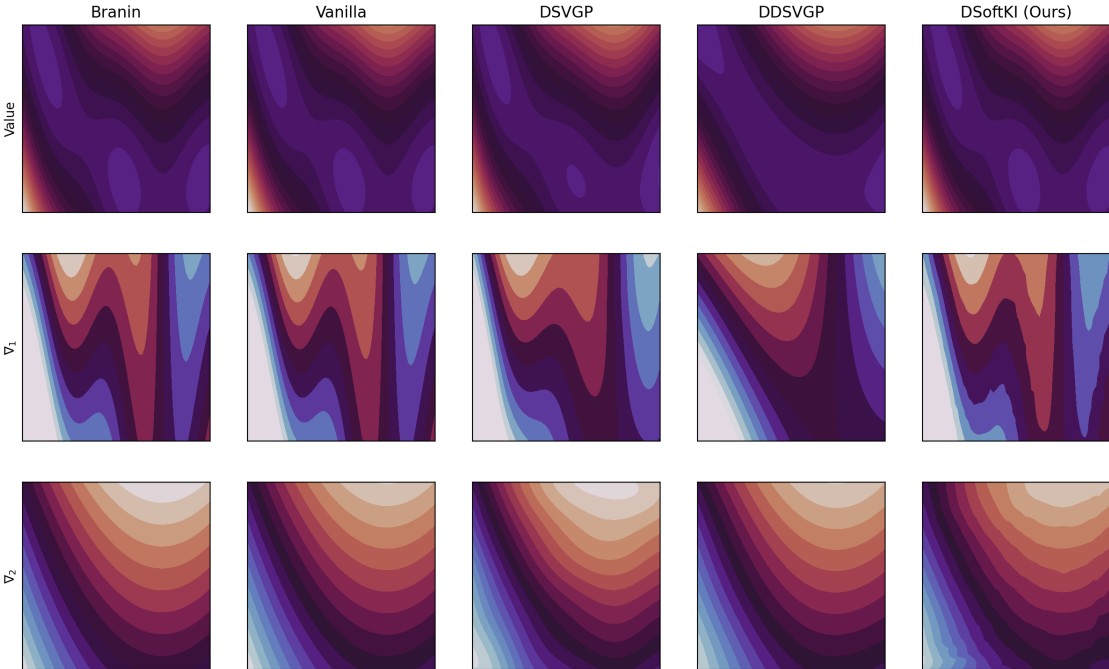

Figure 1: Comparison of GPwD regression with vanilla GPwD, DSVGP, DDSVGP, and DSoftKI on the Branin surface (2D). We plot the contours of the surface, the gradient with respect to the first argument ($\nabla_1$), and the gradient with respect to the second argument ($\nabla_2$). Vanilla GPwD is accurate but intractable for sizable $n$ and/or $d$. DSVGP forms a nice approximation of the original surface but is intractable for large $d$. DDSVGP uses $p = 2$ directional derivatives to enhance scalability but loses fidelity in modeling the surface. DSoftKI provides an accurate and scalable approximation of the surface.

point. This allows the model to encode local directional sensitivity (Section 4.1), enabling the construction of a scalable approximate kernel, including its first and second-order derivatives, through interpolation. Notably, this design eliminates the need for kernel derivatives, facilitating extensions such as Deep Kernel Learning (DKL) (Wilson et al., 2016) which use learned kernels (see Appendix D). This contrasts with methods such as DSVGP/DDSVP, which introduce separate inducing points for values and gradients as well as require computing kernel derivatives, leading to scaling challenges in both $n$ and $d$ (see Appendix E for more details). As a result, DSoftKI achieves a time complexity of $\mathcal{O}(m^2nd)$ for posterior inference. Because each datapoint consists of $d + 1$ values to fit, we can view DSoftKI as achieving similar complexity to an approximate GP (*e.g.*, (Titsias, 2009)) that is fit to $n(d + 1)$ datapoints.

We evaluate DSoftKI on synthetic functions with known derivatives as a baseline to compare against existing GP regression and GPwD regression methods. Since DSoftKI enables derivative modeling with GPs at larger $n$ and $d$ than previously possible, we also evaluate its efficacy on a high-dimensional molecular force field modeling task ($d$ around $100 - 1000$) which requires predicting gradients. To test a broader range of datasets, we also validate performance on a toy n-body physics simulation and standard UCI regression benchmarks with synthetic derivatives (Appendix C). Our experiments show that DSoftKI is a promising method for GPwD regression with full derivative prediction, both in terms of accuracy (*e.g.*, see Figure 1) and scaling to larger $n$ and $d$ than previously possible (Section 5).

## 2 Related Work

One strategy for scaling GPwD regression is to extend an existing scalable GP method to the setting with derivatives. In this vein, both Stochastic Variational Gaussian Process (SVGP) (Hensman et al., 2013) and Structured Kernel Interpolation (SKI) (Wilson & Nickisch, 2015) have been extended to the setting with

| Method | Predict $\nabla$? | Kernel $\nabla$? | Posterior Inference |
|---|---|---|---|
| Vanilla | yes | yes | $\mathcal{O}(n^3 d^3)$ |
| DSVGP | yes | yes | $\mathcal{O}(m^3 d^3)$ |
| DDSVGP | no | yes | $\mathcal{O}(m^3 p^3)$ |
| DSoftKI (ours) | yes | no | $\mathcal{O}(m^2 n d)$ |

Table 1: A comparison of GPwD methods across several dimensions. As a reminder, $n$ is the number of points, $d$ is the dimensionality of the data, $m$ is the number of inducing/interpolation points, and $p$ is the number of inducing directions. The proposed method, DSoftKI, is scalable, predicts full derivative observations, and does not require computing first or second-order derivatives of the kernel to construct the GPwD kernel. The kernel approximations made by each method are discussed in more detail in Appendix A.1.

derivatives. Since we also follow this strategy, we briefly summarize these methods here and refer the reader to Appendix E for more background on these related works.

DSVGP (Padidar et al., 2021) extends SVGP to the setting with derivatives by modifying the SVGP variational approximation to the GPwD kernel, resulting in a kernel matrix of size $m(d+1) \times m(d+1)$ where $m$ is the number of *inducing points* (Snelson & Ghahramani, 2005; Quinonero-Candela & Rasmussen, 2005). Like SVGP, the inducing points in DSVGP are learned by optimizing an evidence lower bound (ELBO) that can be computed with stochastic variational inference. The time complexity of computing the ELBO for each minibatch of optimization has time complexity of $\mathcal{O}(m^3 d^3)$ and the time complexity of posterior inference is $\mathcal{O}(m^3 d^3)$. Since this can be prohibitive for large $d$, DDSVGP (Padidar et al., 2021) utilizes $p \ll d$ directional derivatives to further improve the time complexity of computing the ELBO per minibatch and posterior inference to $\mathcal{O}(m^3 p^3)$ since the $d$-dimensional gradients have been reduced to $p$-dimensional directional derivatives.

DSKI (Eriksson et al., 2018) extends SKI (Wilson & Nickisch, 2015) to the setting with derivatives by approximating the gradients of the SKI interpolation kernel. Like SKI, the resulting DSKI kernel has structure that enables fast matrix-vector multiplications (MVMs), and consequently, conjugate gradient (CG) methods to perform GP inference. Unlike SKI which uses a cubic interpolation scheme, DSKI uses a quintic interpolation scheme to better handle gradient observations, resulting in a time complexity of a single MVM of $O(nd6^d + m \log m)$. Thus, the scaling in the dimension $d$ is even worse than SKI, limiting the application of DSKI to even smaller dimensions. There are other kernel interpolation methods that improve the dimensionality scaling (*e.g.*, see Kapoor et al. (2021); Yadav et al. (2023)) that can be used besides SKI. However, to the best of our knowledge, these works have not been extended to the setting with derivatives.

Beyond the two works above, there has been relatively little additional work done in GPwD regression compared to GP regression. De Roos et al. (2021) tackles GPwD regression in high dimensions but is limited to low $n$ settings. GPyTorch (Gardner et al., 2021) opens the possibility for scalable GP inference on GPUs and provides support for many standard kernels by hard-coding their first and second-order derivatives but does not contain a standard implementation of a scalable GPwD. GP regressions have been scaled with CG solvers (Wang et al., 2019) on multi-GPU hardware and stochastic gradient descent (Lin et al., 2023). However, to the best of our knowledge, this has not been studied in the setting with derivatives. Our work makes significant improvements in the size of $n$ and $d$ that can be handled in GPwD regression with full derivative observations.

## 3 Background

In this section, we review background on GPs (Section 3.1) and GPwDs (Section 3.2). We also review SoftKI since our method extends it to the setting with derivatives (Section 3.3). We begin by introducing notation that will be used throughout this paper.

**Notation.** The notation $\mathbf{A} = [g(i,j)]_{ij}$ defines a matrix whose $i$-th and $j$-th entry $\mathbf{A}_{ij} = g(i,j)$ for some function $g$ defined on indices $i$ and $j$. Given a list of vectors $(x_i \in \mathbb{R}^d)_{i=1}^n$ indexed by $i$, define the vector

$\mathbf{x} = [x_i]_{i1}$ (*i.e.*, $g(i, 1) = x_i$) to be the flattened length $nd$ column vector. Similarly, $\mathbf{x}^T = [x_i]_{1i}$ is the equivalent row vector. The notation $f(\mathbf{x}) = [f(x_i)]_{i1}$ maps a function $f$ over a flattened vector $\mathbf{x}$. Given lists of vectors $(x_i \in \mathbb{R}^d)_{i=1}^n$ and $(x_j' \in \mathbb{R}^d)_{j=1}^m$, define the matrix $\mathbf{K_{xx'}} = [k(x_i, x_j')]_{ij}$ so that it maps a function $k$ over the pair of $\mathbf{x} = [x_i]_{i1}$ and $\mathbf{x}' = [x_i']_{i1}$.

## 3.1 Gaussian Processes

A *Gaussian process* (GP) is a random (continuous) function $f : \mathbb{R}^d \to \mathbb{R}^r$. It is defined by a *mean function* $\mu : \mathbb{R}^d \to \mathbb{R}^r$ and a positive semi-definite function $k : \mathbb{R}^d \times \mathbb{R}^d \to \mathbb{R}^{(r \times r)}$ called a *kernel function*. A GP has Gaussian finite-dimensional distributions so that $f(\mathbf{x}) \sim \mathcal{N}(\mu_{\mathbf{x}}, \mathbf{K_{xx}})$ for any $\mathbf{x}$ where $\mathcal{N}(\mu_{\mathbf{x}}, \mathbf{K_{xx}})$ indicates a Gaussian distribution with mean $\mu_{\mathbf{x}} = \mu(\mathbf{x})$ and *covariance matrix* $\mathbf{K_{xx}} = [k(x_i, x_j)]_{ij}$. Without loss of generality, we will assume $\mu_{\mathbf{x}} = 0$ since we can shift the mean of a Gaussian.

To perform GP regression on the labeled dataset $\{(x_i, y_i) : x_i \in \mathbb{R}^d, y_i \in \mathbb{R}\}_{i=1}^n$, we use the generative process

$$f(\mathbf{x}) \sim \mathcal{N}(0, \mathbf{K_{xx}}) \tag{GP}$$

$$\mathbf{y} \mid f(\mathbf{x}) \sim \mathcal{N}(f(\mathbf{x}), \mathbf{\Lambda}) \tag{likelihood}$$

where $f$ is a GP and $\mathbf{y}$ is $f(\mathbf{x})$ perturbed by Gaussian noise with covariance $\mathbf{\Lambda} = \beta^2 \mathbf{I}$. The noise $\beta^2$ is an example of a GP *hyperparameter*. Others include the kernel *lengthscale* $\ell$ and *scale* $\gamma$. The hyperparameters can be set by maximizing the *marginal log likelihood* (MLL) of a GP

$$\log p(\mathbf{y} \mid \mathbf{x}; \theta) = \mathcal{N}(\mathbf{y} \mid \mathbf{0}, \mathbf{K_{xx}}(\ell, \gamma) + \mathbf{\Lambda}(\beta)) \tag{1}$$

where we have explicitly indicated the dependence of $\mathbf{K_{xx}}$ on $\ell$ and $\gamma$, and $\mathbf{\Lambda}$ on $\beta^2$, for hyperparameters $\theta = (\ell, \gamma, \beta)$. The time complexity of computing the MLL is $O(n^3)$.

Once the hyperparameters have been set, we can perform posterior inference. The posterior predictive distribution has the closed-form solution (Rasmussen & Williams, 2005)

$$p(f(*) \mid \mathbf{x}, \mathbf{y}) = \mathcal{N}(f(*) \mid \mathbf{K_{*x}}(\mathbf{K_{xx}} + \mathbf{\Lambda})^{-1}\mathbf{y}, \mathbf{K_{**}} - \mathbf{K_{*x}}(\mathbf{K_{xx}} + \mathbf{\Lambda})^{-1}\mathbf{K_{x*}}) \tag{2}$$

where $\mathcal{N}(\cdot \mid \mu, \mathbf{\Gamma})$ is notation for the probability density function (pdf) of a Gaussian distribution with mean $\mu$ and covariance $\mathbf{\Gamma}$. The time complexity of posterior inference is $O(n^3)$ which is the complexity of solving the system of linear equations in $n$ variables $(\mathbf{K_{xx}} + \mathbf{\Lambda})\alpha = \mathbf{y}$ for $\alpha$, *i.e.*, $\alpha = (\mathbf{K_{xx}} + \mathbf{\Lambda})^{-1}\mathbf{y}$.

## 3.2 Gaussian Processes with Derivative Information

If $f : \mathbb{R}^d \to \mathbb{R}$ is a GP with kernel $k : \mathbb{R}^d \times \mathbb{R}^d \to \mathbb{R}$, then $\nabla f : \mathbb{R}^d \to \mathbb{R}^d$ is also a GP with kernel $k' : \mathbb{R}^d \times \mathbb{R}^d \to \mathbb{R}^{(d \times d)}$ defined as

$$k'(x, y) = \left[ \frac{\partial^2 k(x, y)}{\partial x_i \partial y_j} \right]_{ij} = \nabla_x k(x, y) \nabla_y^T . \tag{3}$$

Consequently, we can construct a *GP with derivative observations* (GPwD), a random vector-valued function $\tilde{f} : \mathbb{R}^d \to \mathbb{R}^{d+1}$ defined as

$$\tilde{f}(x) = \begin{pmatrix} f(x) \\ \nabla f(x) \end{pmatrix} \tag{4}$$

that simultaneously models a function $f$ and its gradient $\nabla f$. It has a jointly Gaussian distribution $\tilde{f}(\mathbf{x}) \sim \mathcal{N}(0, \tilde{\mathbf{K}}_{\mathbf{xx}})$ where $\tilde{\mathbf{K}}_{\mathbf{xx}} = [\tilde{k}(x_i, x_j)]_{ij}$ and

$$\tilde{k}(x, x') = \begin{pmatrix} k(x, x') & [\frac{\partial k(x, x')}{\partial (x')^T_j}]_{1j} \\ [\frac{\partial k(x, x')}{\partial x_i}]_{i1} & [\frac{\partial^2 k(x, x')}{\partial x_i \partial (x')^T_j}]_{ij} \end{pmatrix} = \begin{pmatrix} \mathbb{I} \\ \nabla_x \end{pmatrix} k(x, x') \begin{pmatrix} \mathbb{I} & \nabla_{x'}^T \end{pmatrix} \tag{5}$$

where $\mathbb{I}$ is the identity operator so that $\tilde{\mathbf{K}}_{\mathbf{xx}}$ is a $n(d+1) \times n(d+1)$ matrix.

To perform GPwD regression on the labeled dataset $\{(x_i, y_i, dy_i) : x_i \in \mathbb{R}^d, y_i \in \mathbb{R}, dy_i \in \mathbb{R}^d\}_{i=1}^n$ where each $dy_i$ is a gradient label, we use the generative process

$$\tilde{f}(\mathbf{x}) \sim \mathcal{N}(0, \tilde{\mathbf{K}}_{\mathbf{xx}}) \qquad \text{(GPwD)}$$

$$\tilde{\mathbf{y}} \mid \tilde{f}(\mathbf{x}) \sim \mathcal{N}(\tilde{f}(\mathbf{x}), \tilde{\mathbf{\Lambda}}) \qquad \text{(likelihood)}$$

where $\tilde{\mathbf{y}} = [y_i \, dy_i^\top]_{1i}^\top$ is a $n(d+1) \times 1$ vector of values and gradient labels and

$$\tilde{\mathbf{\Lambda}} = \left[ \begin{pmatrix} \beta_v^2 & 0 \\ 0 & \beta_g^2 \mathbf{I} \end{pmatrix} \right]_{ij} \tag{6}$$

is a $n(d+1) \times n(d+1)$ diagonal matrix of noises, $\beta_v^2$ for the function value and $\beta_g^2$ for the gradients. A GPwD's hyperparameters can also be set by maximizing the MLL

$$\log p(\mathbf{y} \mid \mathbf{x}; \theta) = \mathcal{N}(\mathbf{y} \mid 0, \tilde{\mathbf{K}}_{\mathbf{xx}}(\ell, \gamma) + \tilde{\mathbf{\Lambda}}(\beta_v, \beta_g)) \tag{7}$$

where $\theta = (\ell, \gamma, \beta_v, \beta_g)$. The time complexity of computing the MLL is $O(n^3 d^3)$.

The posterior predictive distribution is

$$p(\tilde{f}(*) \mid \mathbf{x}, \tilde{\mathbf{y}}) = \mathcal{N}(\tilde{f}(*) \mid \tilde{\mathbf{K}}_{*\mathbf{x}}(\tilde{\mathbf{K}}_{\mathbf{xx}} + \tilde{\mathbf{\Lambda}})^{-1}\tilde{\mathbf{y}}, \tilde{\mathbf{K}}_{**} - \tilde{\mathbf{K}}_{*\mathbf{x}}(\tilde{\mathbf{K}}_{\mathbf{xx}} + \tilde{\mathbf{\Lambda}})^{-1}\tilde{\mathbf{K}}_{\mathbf{x}*}) \tag{8}$$

which takes on the same form as GP regression where we replace the corresponding variable with its $\tilde{(\cdot)}$ version. The complexity of posterior inference is $O(n^3 d^3)$ which is the complexity of solving a system of linear equations in $n(d+1)$ variables. Notably, GPwD regression has an asymptotic dependence on the data dimensionality $d$ since each partial derivative contributes an additional linear equation.

### 3.3 Soft Kernel Interpolation

*Soft kernel interpolation* (SoftKI) (Camaño & Huang, 2025) is an approximate GP method that uses an approximate kernel

$$\mathbf{K}_{\mathbf{xx}}^{\text{SoftKI}} = \mathbf{\Sigma}_{\mathbf{xz}} \mathbf{K}_{\mathbf{zz}} \mathbf{\Sigma}_{\mathbf{zx}} \tag{9}$$

where $\mathbf{\Sigma}_{\mathbf{xz}} = [\sigma_{\mathbf{z}}^j(x_i)]_{ij}$,

$$\sigma_{\mathbf{z}}^j(x) = \frac{\exp\left(-\|x \oslash T - z_j\|\right)}{\sum_{k=1}^m \exp\left(-\|x \oslash T - z_k\|\right)} \tag{10}$$

performs softmax interpolation between $m \ll n$ *interpolation points* $\mathbf{z}$ whose locations are learned, $\oslash$ is a Hadamard division (*i.e.*, element-wise division), and $T \in \mathbb{R}^d$ is a learnable temperature vector akin to using automatic relevance detection (ARD) (MacKay et al., 1994) to set lengthscales for different dimensions. Thus, it is a method that combines aspects of kernel interpolation from SKI and adaptability of inducing point locations to data as in variational inducing points method to obtain a scalable GP method.

Although the interpolation points bear resemblance to inducing points, they are not associated with corresponding inducing variables that have normal priors which are introduced for the purposes of variational optimization. Instead, interpolation points are selected based on distances to data, and so encode the geometry of the underlying data that is most useful for interpolation (Figure 2). They are learned by optimizing a combination of the SoftKI MLL, and an approximate MLL when numeric instability arises due to the current placement of interpolation points. More concretely, the objective is

$$\log \hat{p}(\mathbf{y} \mid \mathbf{x}; \theta) = \begin{cases} \log p(\mathbf{y} \mid \mathbf{x}; \theta) & \text{when numerically stable} \\ \log \bar{p}(\mathbf{y} \mid \mathbf{x}; \theta) & \text{otherwise} \end{cases} \tag{11}$$

where $\log p(\mathbf{y} \mid \mathbf{x}; \theta)$ is the SoftKI MLL and $\log \bar{p}(\mathbf{y} \mid \mathbf{x}; \theta)$ is an approximate MLL termed *Hutchinson's pseudoloss* (Maddox et al., 2022). It is defined as

$$\log \bar{p}(\mathbf{y} \mid \mathbf{x}; \theta) = -\frac{1}{2} \left[ \mathbf{u}_0^\top \mathbf{D}_\theta \mathbf{u}_0 - \frac{1}{l} \sum_{j=1}^l \mathbf{u}_j^\top (\mathbf{D}_\theta \mathbf{w}_j) \right] \tag{12}$$

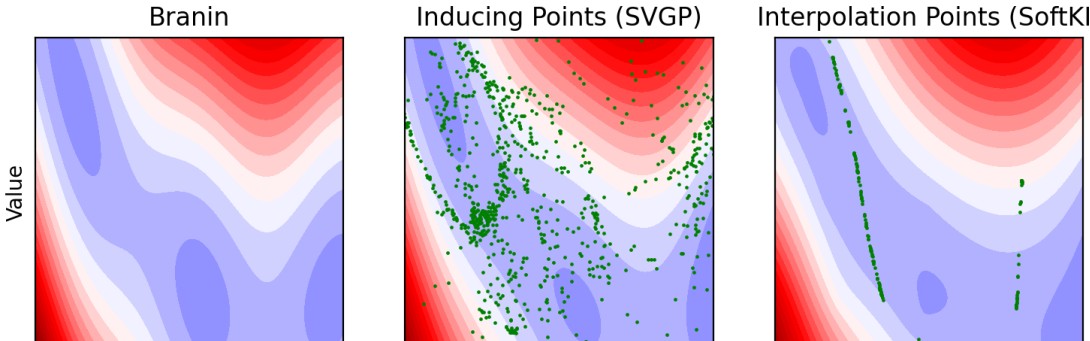

Figure 2: Comparison of learned interpolation points locations versus learned inducing point locations (only those in unit square shown) on the Branin surface. Interpolation point locations encode geometric structure in the data that is most useful for interpolation whereas inducing point locations reflect the normal priors placed on their associated inducing variables.

where $\mathbf{D}_\theta = \mathbf{K}_{\mathbf{xx}}^{\text{SoftKI}}(\ell, \gamma, T, \mathbf{z}) + \mathbf{\Lambda}(\beta)$, $\mathbf{u}_1, \ldots, \mathbf{u}_l$ are solutions to the equations $\mathbf{D}_\theta(\mathbf{u}_0\, \mathbf{u}_1 \ldots \mathbf{u}_l) = (\mathbf{y}\, \mathbf{w}_1 \ldots \mathbf{w}_l)$, and $\mathbf{w}_j$ for $1 \leq j \leq l$ are Gaussian random vectors normalized to have unit length. Observe that

$$\nabla_\theta \log p(\mathbf{y} \,|\, \mathbf{x}; \theta) = -\frac{1}{2}\left[ -\mathbf{u}_0^\top \frac{\partial \mathbf{D}_\theta}{\partial \theta} \mathbf{u}_0 + \text{tr}\left( \mathbf{D}_\theta^{-1} \frac{\partial \mathbf{D}_\theta}{\partial \theta} \right) \right] \tag{13}$$

$$\approx -\frac{1}{2}\left[ -\mathbf{u}_0^\top \frac{\partial \mathbf{D}_\theta}{\partial \theta} \mathbf{u}_0 + \frac{1}{l}\sum_{j=1}^{l} \mathbf{u}_j^\top \frac{\partial \mathbf{D}_\theta}{\partial \theta} \mathbf{w}_j \right] \tag{14}$$

$$= \nabla_\theta \log \tilde{p}(\mathbf{y} \,|\, \mathbf{x}; \theta) \tag{15}$$

so that the gradient of the Hutchinson's pseudoloss is approximately equal to the gradient of the MLL when approximated with Hutchinson's stochastic trace estimator (Girard, 1989; Hutchinson, 1989). The time complexity of computing the Hutchinson's pseudoloss per minibatch of size $b$ is $\mathcal{O}(b^2 + m^3)$.

The posterior predictive distribution is

$$p(f(*) \,|\, y) = \mathcal{N}(\hat{\mathbf{K}}_{*\mathbf{z}}\hat{\mathbf{C}}^{-1}\hat{\mathbf{K}}_{\mathbf{zx}}\mathbf{\Lambda}^{-1}\mathbf{y}, \mathbf{K}_{**}^{\text{SoftKI}} - \mathbf{K}_{*\mathbf{x}}^{\text{SoftKI}}(\mathbf{\Lambda}^{-1} - \mathbf{\Lambda}^{-1}\hat{\mathbf{K}}_{\mathbf{xz}}\hat{\mathbf{C}}^{-1}\hat{\mathbf{K}}_{\mathbf{zx}}\mathbf{\Lambda}^{-1})\mathbf{K}_{\mathbf{x}*}^{\text{SoftKI}}) \tag{16}$$

where $\hat{\mathbf{K}}_{\mathbf{xz}} = \mathbf{\Sigma}_{\mathbf{xz}}\mathbf{K}_{\mathbf{zz}}$ and $\hat{\mathbf{C}} = \mathbf{K}_{\mathbf{zz}} + \hat{\mathbf{K}}_{\mathbf{zx}}\mathbf{\Lambda}^{-1}\hat{\mathbf{K}}_{\mathbf{xz}}$. It is similar to the posterior predictive of a Sparse Gaussian Process Regression (SGPR) (Titsias, 2009), the difference being that we replace kernels in the SGPR posterior with the interpolated versions. The time complexity of posterior inference is $\mathcal{O}(m^2 n)$.

## 4 Method

In this section, we extend SoftKI to work with derivatives. First, we introduce the DSoftKI kernel (Section 4.1). Next, we discuss posterior inference (Section 4.2). Finally, we discuss the role of value and gradient noise hyperparameters in DSoftKI (Section 4.3).

### 4.1 Soft Kernel Interpolation with Derivatives

The DSoftKI kernel takes the same form as DSKI, *i.e.*,

$$\tilde{\mathbf{K}}_{\mathbf{xx}}^{\text{DSoftKI}} = \tilde{\mathbf{\Sigma}}_{\mathbf{xz}}\mathbf{K}_{\mathbf{zz}}\tilde{\mathbf{\Sigma}}_{\mathbf{zx}} \approx \tilde{\mathbf{K}}_{\mathbf{xx}} \tag{17}$$

where

$$\tilde{\mathbf{\Sigma}}_{\mathbf{xz}} = \left[ \begin{pmatrix} \sigma_{\mathbf{z}}^j(x_i) \\ \nabla \sigma_{\mathbf{z}}^j(x_i) \end{pmatrix} \right]_{ij} = \left[ \begin{pmatrix} \mathbb{I} \\ \nabla_{x_i} \end{pmatrix} \sigma_{\mathbf{z}}^j(x_i) \right]_{ij}. \tag{18}$$

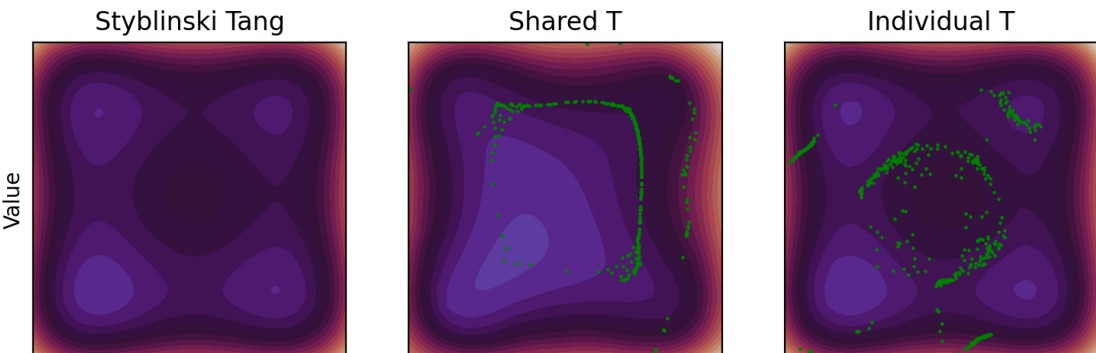

Figure 3: Reconstructed Styblinski Tang surface using a shared temperature (original SoftKI scheme) versus individual temperature (proposed scheme) during interpolation for DSoftKI. We also overlay the learned interpolation points (scaled and translated to fit the unit square) in green to illustrate the differences in learned interpolation points.

Unpacking the definition, we obtain

$$\tilde{\mathbf{K}}_{\mathbf{xx}}^{\text{DSoftKI}} = \left[ \begin{pmatrix} \sigma_{ij}\mathbf{K_{zz}}\sigma_{ij} & \sigma_{ij}\mathbf{K_{zz}}(\partial\sigma_{ij}) \\ (\partial\sigma_{ij})^T\mathbf{K_{zz}}\sigma_{ij} & (\partial\sigma_{ij})^T\mathbf{K_{zz}}(\partial\sigma_{ij}) \end{pmatrix} \right]_{ij} \tag{19}$$

where we have abbreviated $\sigma_{\mathbf{z}}^j(x_i) = \sigma_{ij}$ and $\nabla\sigma_{\mathbf{z}}^j(x_i) = \partial\sigma_{ij}$ to reduce clutter. Notably, the computation of the DSoftKI kernel does not require computing the first-order or second-order derivatives of the kernel. Instead, it is approximated via interpolation. Consequently, it is tractable to perform hyperparameter optimization with first-order gradient methods. Additionally, it can support a wide range of kernels, including learned kernels as in DKL. This contrasts with methods such as DDSVGP where the kernel and its gradients are hard-coded in practice to alleviate the computational cost and to enable tractable hyperparameter optimization with gradient-based methods. We refer the reader to Appendix D for a demonstration of DKL with DSoftKI.

While the DSoftKI kernel affords computational savings, it also increases the burden of kernel interpolation to approximate a kernel's first and second-order derivatives. To solve this challenge, we adapt the softmax interpolation scheme to additionally account for directional orientation of interpolation points relative to the data. More concretely, we replace the global temperature vector with local temperature vectors associated with each interpolation point. In this way, interpolation points that are close to each other can nevertheless model variations in surface curvature as influenced by relevant gradient observations. The original softmax interpolation scheme would be unable to distinguish these variations since the global temperature vector enforces a shared orientation for every interpolation point.

To implement the above idea, we associate each interpolation point $z_k$ with a corresponding learnable temperature vector $T_k \in \mathbb{R}^d$ for $1 \leq k \leq m$ as in

$$\sigma_{\mathbf{z}}^j(x) = \frac{\exp\left(-\|x \oslash T_j - z_j\|\right)}{\sum_{k=1}^m \exp\left(-\|x \oslash T_k - z_k\|\right)}. \tag{20}$$

This introduces $d \times m$ extra parameters $\mathbf{T}^{(d \times m)} = (T_1 \dots T_m)$ to the DSoftKI model that can be learned via hyperparameter optimization. The gradient of the softmax interpolation has closed-form solution

$$\nabla_{x_i}\sigma_{\mathbf{z}}^j(x_i) = -\sum_{k=1}^m \sigma_{\mathbf{z}}^j(x_i)(\delta_{jk} - \sigma_{\mathbf{z}}^k(x_i))\frac{x_i \oslash T_k - z_k}{\|x_i \oslash T_k - z_k\|} \oslash T_k \tag{21}$$

which can be obtained in the standard way via the chain-rule. The $j$-th component of the term

$$\left[\frac{x_i \oslash T_k - z_k}{(\|x_i \oslash T_k - z_k\| + \epsilon)} \oslash T_k\right]_j = \frac{x_{ij} - T_{kj}z_{kj}}{T_{kj}^2(\|x_i \oslash T_k - z_k\| + \epsilon)} \tag{22}$$

---

**Algorithm 1** DSoftKI regression adapts the SoftKI algorithm to handle the additional computational challenges of the DSoftKI kernel and learning local temperature vectors $\mathbf{T}^{(m \times d)}$.

---

**Require:** DSoftKI hyperparameters $\theta = (\ell, \gamma, \mathbf{z}^{(m \times d)}, \mathbf{T}^{(m \times d)}, \beta_v, \beta_g)$.
**Require:** Dataset $(\mathbf{x}, \tilde{\mathbf{y}})$.
**Require:** Optimization hyperparameters: batch size $b$, number of epochs $E$, and learning rate $\eta$.
**Ensure:** Learned DSoftKI coefficients $\alpha$.

 1: **for** $i = 1$ to E **do**
 2:     **for** $\mathbf{x}_b, \tilde{\mathbf{y}}_b$ in $\texttt{batch}((\mathbf{x}, \tilde{\mathbf{y}}), b)$ **do**                   ▷ $\texttt{batch}$ splits the dataset into chunks of size $b$
 3:         $\mathbf{L}, \mathbf{L}^T \leftarrow \texttt{Cholesky}(\mathbf{K}_{\mathbf{zz}})$
 4:         $\mathbf{F} \leftarrow \tilde{\mathbf{\Sigma}}_{\mathbf{x}_b \mathbf{z}} \mathbf{L}$                 ▷ Low rank representation of $\tilde{\mathbf{K}}_{\mathbf{x}_b \mathbf{x}_b}^{\mathrm{DSoftKI}}(\ell, \gamma, \mathbf{z}, \mathbf{T}) = \mathbf{F}\mathbf{F}^T$
 5:         $\tilde{\mathbf{D}}_\theta \leftarrow [\![\mathbf{F}\mathbf{F}^T + \tilde{\mathbf{\Lambda}}(\beta_v, \beta_g)]\!]_z$          ▷ $[\![\cdot]\!]_z$ delays computation of result until needed
 6:         $\theta \leftarrow \theta + \eta \nabla_\theta \log \hat{\tilde{p}}(\tilde{\mathbf{y}}_b \,|\, \mathbf{x}_b; \tilde{\mathbf{D}}_\theta)$        ▷ Stabilized DSoftKI MLL (Equation 25)
 7:     **end for**
 8: **end for**
 9: $\alpha \leftarrow \texttt{solve}(\hat{\tilde{\mathbf{C}}}\alpha = \hat{\tilde{\mathbf{K}}}_{\mathbf{zx}} \tilde{\mathbf{\Lambda}}^{-1} \tilde{\mathbf{y}})$           ▷ Solve system of linear equations (Appendix A.2)
10: **return** $\alpha$

---

where we have added a small factor $\epsilon > 0$ to the denominator to avoid division by zero, is proportional to the direction from the scaled interpolation point $T_k \odot z_k$ to the data point $x_i$ along dimension $j$, with scaling factor $1/T_{kj}^2$. As a result, points that are nearby can nevertheless have different influences on the interpolation strength of the gradients depending on their respective directions. Figure 3 illustrates the differences in surface reconstruction using a shared versus individual temperature vector across interpolation points. We refer the reader to Appendix B.5 for further discussion on the role of temperatures and their connection with lengthscales.

## 4.2 Posterior Inference

Algorithm 1 summarizes the adaptation of the SoftKI algorithm to work with the DSoftKI kernel. Since DSoftKI introduces a different interpolation scheme and more learnable parameters, this introduces different optimization dynamics and computational concerns.

**Hyperparameter optimization.** We learn the locations of the interpolation points by using stochastic gradient descent on the stabilized DSoftKI MLL. The MLL of DSoftKI is

$$\log p(\tilde{\mathbf{y}}|\mathbf{x}; \theta) = \mathcal{N}(\tilde{\mathbf{y}} \,|\, 0, \tilde{\mathbf{D}}_\theta). \tag{23}$$

where $\tilde{\mathbf{D}}_\theta = \tilde{\mathbf{K}}_{\mathbf{xx}}^{\mathrm{DSoftKI}}(\ell, \gamma, \mathbf{z}, \mathbf{T}) + \tilde{\mathbf{\Lambda}}(\beta_v, \beta_g)$ for $\theta = (\ell, \gamma, \mathbf{z}, \mathbf{T}, \beta_v, \beta_g)$. We refer the reader to Appendix B.1 for details on hyperparameter initialization. For a minibatch of size $b$, the resulting matrix is of size $b(d + 1) \times b(d + 1)$ as opposed to $b \times b$ in SoftKI. For sizable $b$ and $d$, this would be intractable to compute with since it requires solving a system of linear equations.

As a result, we decompose $\mathbf{K}_{\mathbf{xx}}^{\mathrm{DSoftKI}} = \mathbf{F}\mathbf{F}^T$ where $\mathbf{K}_{\mathbf{zz}} = \mathbf{L}\mathbf{L}^T$ is a Cholesky decomposition and $\mathbf{F} = \tilde{\mathbf{\Sigma}}_{\mathbf{xz}} \mathbf{L}$. This changes the space requirement from $\mathcal{O}(b^2 d^2)$ for a direct representation to $\mathcal{O}(bmd)$. In practice, we expect $m < bd$, since $m$ and $b$ are around the same order of magnitude. Since the Cholesky decomposition is necessary to retain a tractable representation of the DSoftKI kernel, we perform it in double-precision when we encounter numerical instability. In our experience, we rarely encounter numerical instability due to the current placement of interpolation points in DSoftKI. We believe that this is the case since interpolation points can additionally take into account directional information in DSoftKI so that points that are close by distance-wise can nevertheless be oriented to prioritize different directions based on gradient observations.

The factored representation is used to compute the DSoftKI MLL using a lazy representation

$$\tilde{\mathbf{D}}_\theta = [\![\mathbf{F}\mathbf{F}^T + \tilde{\mathbf{\Lambda}}(\beta_v, \beta_g)]\!]_z \tag{24}$$

that directly stores $\mathbf{F}$ and $\tilde{\mathbf{\Lambda}}(\beta_v, \beta_g)$, and $[\![\cdot]\!]_z$ delays computation of any intermediate result until needed. To take advantage of this representation, we use a low-rank multivariate Gaussian distribution which can

| | d | SoftKI | SVGP | DSVGP | DDSVGP | DSoftKI |
|---|---|---|---|---|---|---|
| Branin | 2 | $0.004 \pm 0.0$ | $0.018 \pm 0.004$ | $0.088 \pm 0.062$ | $0.176 \pm 0.031$ | $\mathbf{0.003 \pm 0.001}$ |
| Six-hump-camel | 2 | $0.026 \pm 0.003$ | $0.05 \pm 0.003$ | $0.101 \pm 0.031$ | $0.669 \pm 0.04$ | $\mathbf{0.015 \pm 0.006}$ |
| Styblinski-tang | 2 | $0.025 \pm 0.001$ | $0.05 \pm 0.003$ | $0.101 \pm 0.01$ | $0.125 \pm 0.06$ | $\mathbf{0.012 \pm 0.002}$ |
| Hartmann | 6 | $0.05 \pm 0.002$ | $0.164 \pm 0.006$ | $0.335 \pm 0.009$ | $0.346 \pm 0.019$ | $\mathbf{0.011 \pm 0.004}$ |
| Welch | 20 | $0.01 \pm$ nan | $0.065 \pm 0.001$ | - | $0.578 \pm 0.013$ | $\mathbf{0.003 \pm 0.003}$ |

Table 2: Test RMSE (best bolded) on selected synthetic datasets. One of the runs for SoftKI encountered numerical instability (hence $\pm$nan). $-$ indicates a timeout for DSVGP.

avoid working with the full $b(d+1) \times b(d+1)$ matrix $\tilde{\mathbf{D}}_\theta$ via the Woodbury matrix identity and matrix determinant lemma. This reduces the challenge of computing the MLL to computing the determinant and inverses of the $m \times m$ capacitance matrix $\mathbf{I} + \mathbf{F}^T \tilde{\mathbf{\Lambda}}(\beta_v, \beta_g)^{-1}\mathbf{F}$ instead which takes time complexity $\mathcal{O}(m^2 bd)$ to form. When numerical instability is encountered during the computation of DSoftKI's MLL, we compute Hutchinson's pseudoloss. The time complexity of a single MVM is $\mathcal{O}(mbd)$. The stabilized DSoftKI MLL is thus

$$\log \hat{\hat{p}}(\tilde{\mathbf{y}} \mid \mathbf{x}; \theta) = \begin{cases} \log p(\tilde{\mathbf{y}} \mid \mathbf{x}; \theta) & \text{when numerically stable} \\ \log \bar{p}(\tilde{\mathbf{y}} \mid \mathbf{x}; \theta) & \text{otherwise} \end{cases}. \tag{25}$$

Appendix B.2 contains an ablation of DSoftKI using the DSoftKI MLL compared to the Hutchinson's pseudoloss.

**Posterior inference.** Once we have learned the interpolation points, we can construct the posterior predictive distribution. It is SoftKI's posterior, with the corresponding variables replaced with the $(\tilde{\cdot})$ versions

$$p(\tilde{f}(*) \mid \tilde{\mathbf{y}}) = \mathcal{N}(\hat{\tilde{\mathbf{K}}}_{*\mathbf{z}}\hat{\tilde{\mathbf{C}}}^{-1}\hat{\tilde{\mathbf{K}}}_{\mathbf{zx}}\tilde{\mathbf{\Lambda}}^{-1}\tilde{\mathbf{y}}, \tilde{\mathbf{K}}_{**}^{\text{DSoftKI}} - \tilde{\mathbf{K}}_{*\mathbf{x}}^{\text{DSoftKI}}(\tilde{\mathbf{\Lambda}}^{-1} - \tilde{\mathbf{\Lambda}}^{-1}\hat{\tilde{\mathbf{K}}}_{\mathbf{xz}}\hat{\tilde{\mathbf{C}}}^{-1}\hat{\tilde{\mathbf{K}}}_{\mathbf{zx}}\tilde{\mathbf{\Lambda}}^{-1})\tilde{\mathbf{K}}_{\mathbf{x}*}^{\text{DSoftKI}}) \tag{26}$$

where $\hat{\tilde{\mathbf{K}}}_{\mathbf{xz}} = \tilde{\mathbf{\Sigma}}_{\mathbf{xz}}\mathbf{K}_{\mathbf{zz}}$ and $\hat{\tilde{\mathbf{C}}} = \mathbf{K}_{\mathbf{zz}} + \hat{\tilde{\mathbf{K}}}_{\mathbf{zx}}\tilde{\mathbf{\Lambda}}^{-1}\hat{\tilde{\mathbf{K}}}_{\mathbf{xz}}$. Since each data point in GPWD regression introduces $d+1$ values to fit, we have a system of $n(d+1)$ equations in $m$ variables. Consequently, solving the system of linear equations takes $\mathcal{O}(ndm^2)$ time and space. For large $d$, this can exceed GPU memory limits. Consequently, we sometimes solve these equations on a CPU. It would be an interesting direction of future work to see how this can be improved such as with alternative linear solvers or utilizing multi-GPU hardware.

### 4.3 The Role of Value and Gradient Noises

Since DSoftKI forms its approximate kernel and its derivatives via interpolation, the choice of value and gradient noises become intertwined. To examine this further, we can unpack the posterior mean equation and see that

$$\hat{\tilde{\mathbf{C}}}\alpha = \left[\sum_{j=1}^{m} k(z_a, z_j)\omega_j\right]_a \tag{27}$$

where

$$\omega_j = \sum_{i=1}^{n}\left(\frac{1}{\beta_v^2}\sigma_{ij}y_i + \frac{1}{\beta_g^2}(\partial\sigma_{ij})^T dy_i\right) \tag{28}$$

for $1 \le a \le m$. This indicates that each posterior coefficient $\alpha_a$ jointly influences the reconstruction of function values and their gradients as opposed to introducing separate posterior coefficients for values and gradients. The influence of fitting values versus gradients on the weights is determined by the ratio $\beta_g^2/\beta_v^2$. A ratio of $\beta_g^2/\beta_v^2 = d$ suggests that values and gradients are equally weighted because each gradient has $d$ components. A ratio larger than $d$ thus prioritizes gradients while a ratio smaller than $d$ prioritizes values. We investigate the impact of the ratio on the value and gradient test RMSE more in Appendix B.3.

|  | SoftKI | SVGP | DSVGP | DDSVGP | DSoftKI |
|---|---|---|---|---|---|
| Branin | -0.725 ± 0.017 | 0.116 ± 0.0 | -1.044 ± 0.885 | -0.532 ± 0.068 | **-4.432 ± 0.162** |
| Six-hump-camel | -0.743 ± 0.057 | 0.123 ± 0.001 | -1.539 ± 0.159 | -0.64 ± 0.058 | **-2.175 ± 0.237** |
| Styblinski-tang | -0.54 ± 0.082 | 0.123 ± 0.001 | -1.182 ± 0.166 | -1.064 ± 0.188 | **-2.296 ± 0.045** |
| Hartmann | -0.494 ± 0.111 | 0.221 ± 0.001 | -1.362 ± 0.029 | -1.324 ± 0.027 | **-3.16 ± 0.325** |
| Welch | -0.699 ± nan | 0.141 ± 0.0 | - | 0.706 ± 0.055 | **-1.14 ± 4.795** |

Table 3: Test NLL (best bolded) on selected synthetic datasets. One of the runs for SoftKI encountered numerical instability (hence ±nan). − indicates a timeout for DSVGP.

|  | d | SoftKI | SVGP | DSVGP | DDSVGP | DSoftKI |
|---|---|---|---|---|---|---|
| Branin | 2 | 0.194 ± 0.005 | 0.339 ± 0.149 | **1.318 ± 0.02** | 1.945 ± 0.022 | 2.555 ± 0.199 |
| Six-hump-camel | 2 | 0.257 ± 0.092 | 0.373 ± 0.104 | **1.31 ± 0.008** | 1.955 ± 0.062 | 2.434 ± 0.087 |
| Styblinski-tang | 2 | 0.21 ± 0.006 | 0.335 ± 0.14 | **1.316 ± 0.018** | 1.91 ± 0.016 | 2.539 ± 0.137 |
| Hartmann | 6 | 0.199 ± 0.003 | 0.337 ± 0.126 | 9.305 ± 0.113 | **1.934 ± 0.045** | 2.544 ± 0.091 |
| Welch | 20 | 0.213 ± 0.002 | 0.418 ± 0.123 | - | **1.944 ± 0.063** | 4.452 ± 0.192 |

Table 4: Wall-clock training time in seconds per epoch (best GPwD bolded) on selected synthetic dataset. − indicates a timeout for DSVGP.

## 5 Experiments

In this section, we evaluate DSoftKI on synthetic functions (Section 5.1) to obtain a baseline of comparison and high-dimensional molecular force field (Section 5.2) to test scale. We refer the reader to Appendix C for more experiments with a toy n-body simulation and on the UCI dataset (Dua & Graff, 2017) with synthetic gradients.

**Baseline GP and GPwD methods.** We use the default GPyTorch implementations (Gardner et al., 2018) of SVGP, DSVGP, and DDSVGP with 2 inducing directions as baseline variational GPs. We use the PLL (Jankowiak et al., 2020) modification to the ELBO for the variational GPs as recommended by (Padidar et al., 2021) to allow separate noise parameters for function values ($\beta_v^2$) and gradients ($\beta_g^2$). We also use SoftKI as a baseline GP where we also adopt the DSoftKI interpolation scheme. We do not compare against DSKI since the dimensionality of the datasets are too high. We use the RBF kernel (with scale) using automatic relevance determination (ARD) lengthscales with the exception of DDSVGP which does not support it. Unless otherwise stated, we use $m = 512$ inducing points for all methods. We optimize all hyperparameters using the Adam (Kingma & Ba, 2014) optimizer. All GPs and GPwDs use single-precision floating point numbers, except for DSoftKI which uses double-precision occasionally to enhance the stability of Cholesky decomposition as described previously.

### 5.1 Regression with and without Derivative Information

Our first experiment tests DSoftKI on selected synthetic test functions ranging from $d = 2$ to $d = 20$ with known derivatives following (Padidar et al., 2021) so that we can compare against existing GP regression and GPwD regression. For each function, we generate 20000 datapoints, using 10000 points for training and reserving 10000 points for testing.

**Effective learning rate.** Since the training data that each GP observes varies across methods due to how each method utilizes derivative information, setting up a fair comparison between each method is not straightforward. For instance, even comparing the variational methods SVGP, DSVGP, and DDSVGP is nuanced since the amount of training data each method encounters is different at $n$, $n(d + 1)$, and $n(p + 1)$ where $p$ is the number of inducing directions respectively. To control for this, we choose to use the notion of an *effective learning rate* $\Delta_{\text{eff}}$ defined as $\Delta_{\text{eff}} = D\Delta_{\text{base}}$ where $D$ is a method-specific number of derivative

|  | DSoftKI* RMSE | DSoftKI* NLL | Δ RMSE | Δ NLL |
|---|---|---|---|---|
| `Branin` | $0.002 \pm 0.0$ | $-4.266 \pm 0.009$ | $-0.000$ | $0.166$ |
| `Six-hump-camel` | $0.163 \pm 0.02$ | $-0.392 \pm 0.112$ | $0.148$ | $1.783$ |
| `Styblinski-tang` | $0.026 \pm 0.002$ | $-1.829 \pm 0.171$ | $0.014$ | $0.467$ |
| `Hartmann` | $0.336 \pm 0.034$ | $0.336 \pm 0.092$ | $0.326$ | $3.495$ |
| `Welch` | $0.027 \pm 0.008$ | $-1.311 \pm 0.405$ | $0.024$ | $-0.171$ |

Table 5: DSoftKI* uses the original interpolation scheme proposed in SoftKI. Δ RMSE and NLL give the increase in RMSE and NLL respectively compared to the DSoftKI interpolation scheme.

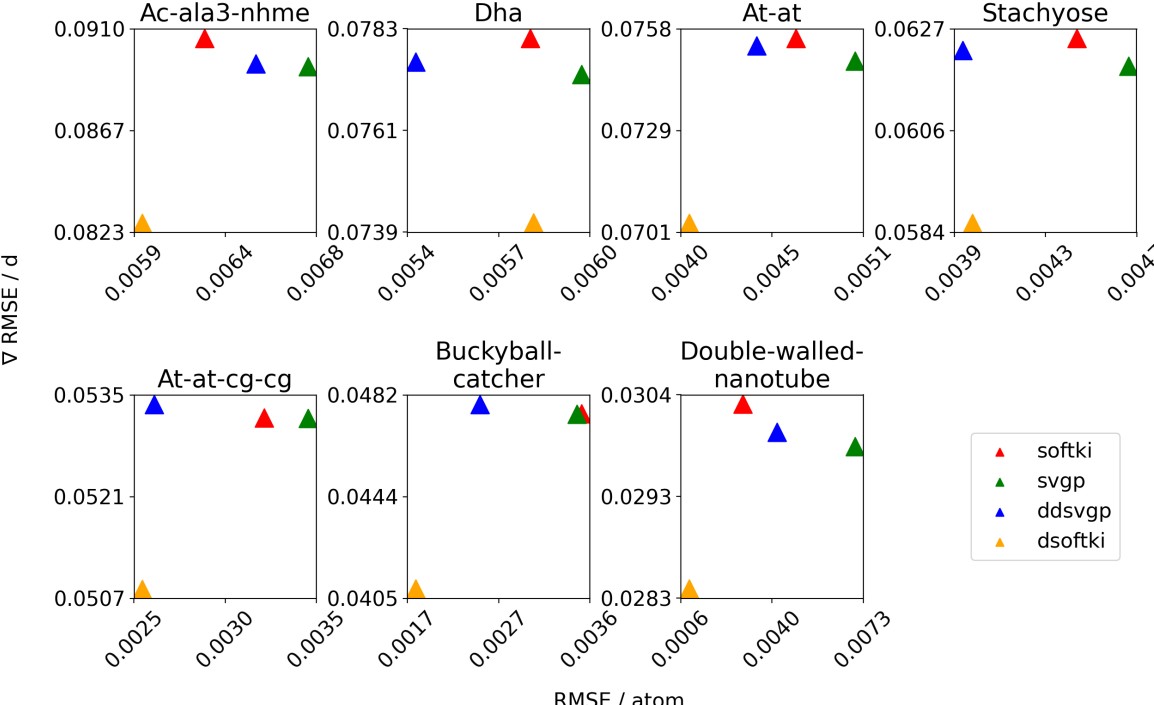

Figure 4: Test RMSE per atom vs. test gradient RMSE per component obtained by various methods on MD22 dataset. Bottom left is best.

dimensions and $\Delta_{\text{base}}$ is a base learning rate. The learning rate that we use for hyperparameter optimization is thus $\Delta_{\text{eff}}$ which we obtain for a constant $\Delta_{\text{base}}$ and minibatch size across methods. In this way, we keep all the training data, epochs, and minibatch size constant across methods while requiring hyperparameter optimization to take a step in proportion to the amount of data that it encounters. We use $D = 1$ for GP regression, $D = d$ for DSVGP, $D = d+1$ for DDSVGP, and $D = \beta_g^2/(d\beta_v^2)+1$ for DSoftKI. We set $\beta_g^2 = d\beta_v^2$ so that $D = 2$ for DSoftKI.

**Results.** Table 2 gives the test root mean-squared error (RMSE) and Table 3 test negative log-likelihood (NLL) averaged across three runs. We use $\Delta_{\text{base}} = 0.01$ and a minibatch size of 1024, settings that are known to work well for SVGP on this benchmark, to set $\Delta_{\text{eff}}$ for all other methods. We were not able to run DSVGP on the Welch dataset due to time constraints since $d = 20$ and each minibatch of hyperparameter optimization has time complexity $\mathcal{O}(m^3 d^3)$.

Once we control for the effective learning rate, we see that fitting derivatives helps improve DSoftKI's performance relative to SoftKI, its non-derivative base, as measured by test RMSE performance. We believe that this increase in performance is due to DSoftKI's modified interpolation scheme which jointly takes into

---

[2]dfdf

|  | n | d | SoftKI | SVGP | DDSVGP | DSoftKI |
|---|---|---|---|---|---|---|
| `Ac-ala3-nhme` | 76598 | 126 | $0.894 \pm 0.006$ | $1.051 \pm 0.018$ | $\mathbf{7.907 \pm 0.004}$ | $82.665 \pm 0.06$ |
| `Dha` | 62777 | 168 | $0.887 \pm 0.021$ | $0.951 \pm 0.027$ | $\mathbf{6.544 \pm 0.012}$ | $89.155 \pm 0.223$ |
| `At-at` | 18000 | 180 | $0.336 \pm 0.006$ | $0.427 \pm 0.001$ | $\mathbf{2.626 \pm 0.01}$ | $27.236 \pm 0.037$ |
| `Stachyose` | 24544 | 261 | $0.475 \pm 0.005$ | $0.555 \pm 0.007$ | $\mathbf{3.344 \pm 0.021}$ | $54.181 \pm 0.142$ |
| `At-at-cg-cg` | 9137 | 354 | $0.355 \pm 0.008$ | $0.423 \pm 0.007$ | $\mathbf{1.988 \pm 0.006}$ | $27.102 \pm 0.036$ |
| `Buckyball-catcher` | 5491 | 444 | $0.244 \pm 0.007$ | $0.295 \pm 0.004$ | $\mathbf{1.249 \pm 0.007}$ | $20.764 \pm 0.111$ |
| `Double-walled`[†] | 4528 | 1110 | $0.409 \pm 0.007$ | $0.434 \pm 0.004$ | $\mathbf{1.906 \pm 0.003}$ | $45.379 \pm 0.007$ |

Table 6: Wall-clock training time in seconds per epoch (best GPwD bolded) on MD22 dataset. [†]We abbreviate `Double-walled-nanotube` (Figure 4) as `Double-walled` due to space.

account distances and directional information via learned temperature hyperparameters. In Table 5, we run DSoftKI with the SoftKI interpolation scheme, and see that it performs worse on several datasets. For the variational methods, we observe that SVGP performs the best, followed by DDSVGP which fits two directions, and followed last by DSVGP which fits all derivative information once effective learning rate is controlled for. As a reminder, we use $\Delta_{\text{eff}}$ to control for the amount of data each method sees during a step of hyperparameter optimization and not as a recommendation to obtain the best possible results. We conjecture that the difference in optimization dynamics between DSoftKI and DSVGP/DDSVGP is because the former jointly models values and gradients with each interpolation point whereas the latter approaches introduce separate inducing points for values and gradients/directions.

Another trend that we observe once we control for $\Delta_{\text{eff}}$ is that GPwDs tend to have lower NLLs compared to their non-derivative counterparts. For methods based on SVGP, this is somewhat surprising since the variants that fit derivatives also have higher test RMSE, indicating that the versions that fit derivatives result in simpler models. On the other hand, for DSoftKI, we observe that the NLL is positively correlated with its test RMSE relative to the performance of SoftKI. This suggests that the additional information is used to increase the complexity of the model when it also improves the fit of the model.

Table 4 reports the (wall-clock) training time (in seconds) per epoch of training on selected synthetic datasets. One epoch of training is one iteration through the dataset. We use a single RTX 6000 Blackwell Pro Max-Q edition GPU. We observe that the non-derivative methods such as SoftKI and SVGP are faster than the derivative-fitting counterparts. This is expected since we fit $d$ times more data in GPwD regression. For methods that predict full derivative observations, we see that DSoftKI has higher per-epoch training time compared to DSVGP, another method that can predict full derivative observations, on datasets with $d = 2$. However, as $d$ scales, DSVGP's cubic scaling in $d$ leads to prohibitive cost and it becomes infeasible to run DSVGP on the `Welch` ($d = 20$) dataset. DSoftKI also has higher per-epoch training time than DDSVGP, particularly for large d. This is expected given DSoftKI's $O(m^2nd)$ complexity versus DDSVGP's $O(m^3p^3)$ with $p = 2$ so that its per-epoch time does not scale with increasing $d$. However, DDSVGP only predicts $p = 2$ directional derivatives.

## 5.2 High-Dimensional Molecular Force Fields

In this section, we evaluate the ability of DSoftKI to fit gradient information. Towards this end, we use the MD22 dataset (Chmiela et al., 2023), a dataset of molecular energy surfaces where inputs are molecular configurations ($d = 168$ to $1110$), outputs are energies, and gradients of the energy surface correspond to the physically-meaningful quantity of (negative) forces. Consequently, fitting this information can enable us to model the physical dynamics.

**Obtaining gradient predictions.** To obtain the gradient prediction for SVGP, SoftKI, and DDSVGP, we take the gradient of the posterior predictive mean to obtain the predicted gradient value since none of these methods predict the gradient directly. For DSoftKI, we use the prediction of the gradients and not the gradient of the prediction. In an exact GPwD, these predictions would be equivalent. We are not able to scale DSVGP to this setting.

**Results.** Figure 4 reports the test value RMSE (per atom) vs. the gradient RMSE (per component) averaged across 3 runs for each method on each molecule. We refer the reader to Appendix B.5 for more experimental details. The bottom left corner is the best, since that is where there is minimum error on both values and gradients. We observe that DSoftKI performs well, particularly when the dimension of the dataset is large. One surprising observation we make is that lower value error is not correlated with lower gradient error. Upon examining the test RMSE curves (see Appendix B.5), we see that many GPs that do not fit full derivative information such as SVGP increase the test RMSE error for derivative prediction as training progresses. Conversely, DSoftKI improves the test RMSE error on both values and gradients as training progresses for most datasets. Further investigation of the tradeoffs of fitting gradients is warranted for GPwD regression, since the additional computational complexity of fitting derivatives should be weighed against if obtaining accurate gradients predictions is required or not.

Table 6 reports the (wall-clock) training time (in seconds) per epoch of training on the MD22 dataset. We include SoftKI and SVGP as non-derivative methods for completeness. As a reminder, DVSGP does not scale to this setting and we use DDSVGP with $p = 2$ inducing directions. As expected, DSoftKI has higher per-epoch training time than DDSVGP, particularly for large d. The additional time complexity is justified in cases such as molecular dynamics where full forces are required since the shape of the surface also needs to be fit. Notably, DSoftKI's training time scales roughly linearly with $d$ (*e.g.*, 20.8 seconds at $d = 444$ versus 45.4 seconds at $d = 1110$), consistent with its $O(m^2nd)$ complexity. In contrast, DDSVGP's time remains nearly constant across dimensions since its $O(m^3p^3)$ complexity is independent of $d$. However, this assumes $mp^2 > d$, which we observe begins to break when comparing its timing on the `Double-walled` dataset with the `Buckyball-catcher` dataset.

## 6 Conclusion

In this paper, we introduce a GP that can fit and predict full derivative information called DSoftKI. It has posterior inference time complexity of $\mathcal{O}(ndm^2)$ and hyperparameter optimization time complexity of $\mathcal{O}(m^3 + bmd)$, and thus, scales to larger $n$ and $d$ than supported by previous GPwD methods while retaining the ability to fit and predict full derivative information. We have evaluated DSoftKI on GPwD regression tasks and shown that it accurate, both in terms of test RMSE and test NLL, as well as modeling gradients. While promising, there are certain limitations of our work that should be investigated further.

First, while DSoftKI is more scalable than existing GPwD methods that fit derivatives, further work can be done to improve its computational complexity. In particular, the method is currently bottlenecked by the computation of the DSoftKI MLL. Improved approximations of the MLL or a different objective could further scale the efficiency of fitting derivative observations. Second, a deeper investigation of the tradeoffs between fitting derivative information or not when they are available is warranted. For some applications, the added difficulty of fitting gradient observations may not warrant improved surface modeling. Third, while we demonstrate that DSoftKI supports DKL out of the box (Appendix D), achieving computational speedups by projecting into a lower-dimensional feature space, further exploration of learned kernels for GPwD regression is a promising direction.

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

# A Method

We describe the DSoftKI method in more detail, including comparing its kernel structure with related GPwD methods (Section A.1), detailing the solving procedure (Section A.2), and its implementation (Section A.3).

## A.1 Comparing Gradient Kernels

For a closer comparison of exact GPwDs, DDSVGP, DSVGP, and DSoftKI, we write out the kernel matrix between a test point $*$ and the training data $\mathbf{x}$ or inducing points $\mathbf{z}$ in more detail. For an exact GPwD, the kernel directly uses the training data and the derivatives of the kernel are used exactly as below

$$\tilde{\mathbf{K}}_{*\mathbf{x}} = \left[ \begin{pmatrix} \mathbb{I} \\ \nabla_* \end{pmatrix} k(*, x_j) \begin{pmatrix} \mathbb{I} & \nabla_{x_j}^T \end{pmatrix} \right]_{*j} \tag{29}$$

$$= \left[ \begin{pmatrix} k(*, x_j) & k(*, x_j)\nabla_{x_j}^T \\ \nabla_* k(*, x_j) & \nabla_* k(*, x_j)\nabla_{x_j}^T \end{pmatrix} \right]_{*j}. \tag{30}$$

For a DDSVGP (*i.e.*, with directional derivatives), the kernel uses the inducing points and additional variational parameters to approximate the training data. Additionally, it uses directional derivatives of the underlying kernel function as below

$$\bar{\mathbf{K}}_{*\mathbf{z}}^{\text{DDSVGP}} = \left[ \begin{pmatrix} \mathbb{I} \\ \nabla_* \end{pmatrix} k(*, z_j) \begin{pmatrix} \mathbb{I} & \partial_{\mathbf{V}_j}^T \end{pmatrix} \right]_{*j} \tag{31}$$

$$= \left[ \begin{pmatrix} k(*, z_j) & k(*, z_j)\partial_{\mathbf{V}_j}^T \\ \nabla_* k(*, z_j) & \nabla_* k(*, z_j)\partial_{\mathbf{V}_j}^T \end{pmatrix} \right]_{*j}. \tag{32}$$

For a DSVGP, the kernel uses the inducing points, additional variational parameters, and ordinary derivatives of the underlying kernel as below

$$\bar{\mathbf{K}}_{*\mathbf{z}}^{\text{DSVGP}} = \left[ \begin{pmatrix} \mathbb{I} \\ \nabla_* \end{pmatrix} k(*, z_j) \begin{pmatrix} \mathbb{I} & \nabla_{z_j}^T \end{pmatrix} \right]_{*j} \tag{33}$$

$$= \left[ \begin{pmatrix} k(*, z_j) & k(*, z_j)\nabla_{z_j}^T \\ \nabla_* k(*, z_j) & \nabla_* k(*, z_j)\nabla_{z_j}^T \end{pmatrix} \right]_{*j}. \tag{34}$$

For DSoftKI, the kernel uses the inducing points, an interpolation function, and gradients of the interpolation function as below

$$\tilde{\mathbf{K}}_{*\mathbf{z}}^{\text{DSoftKI}} = \left[ \begin{pmatrix} \mathbb{I} \\ \nabla_* \end{pmatrix} \sigma_{\mathbf{z}}^j(*) \right]_{*j} \mathbf{K}_{\mathbf{zz}} \left[ \sigma_{\mathbf{z}}^j(x_i)^T \begin{pmatrix} \mathbb{I} & \nabla_{x_i}^T \end{pmatrix} \right]_{ji} \tag{35}$$

$$= \left[ \begin{pmatrix} \sigma_{\mathbf{z}}^j(*) \, \mathbf{K}_{\mathbf{zz}} \, \sigma_{\mathbf{z}}^j(x_i')^T & \sigma_{\mathbf{z}}^j(*) \, \mathbf{K}_{\mathbf{zz}} \, (\sigma_{\mathbf{z}}^j(x_i')^T \nabla_{x_i'}^T) \\ (\nabla_* \sigma_{\mathbf{z}}^j(*)) \, \mathbf{K}_{\mathbf{zz}} \, \sigma_{\mathbf{z}}^j(x_i')^T & (\nabla_* \sigma_{\mathbf{z}}^j(*)) \, \mathbf{K}_{\mathbf{zz}} \, (\sigma_{\mathbf{z}}^j(x_i')^T \nabla_{x_i'}^T) \end{pmatrix} \right]_{*i'}. \tag{36}$$

### A.2 Solving

The procedure

$$\alpha \leftarrow \texttt{solve}(\hat{\tilde{\mathbf{C}}}\alpha = \hat{\tilde{\mathbf{K}}}_{\mathbf{zx}} \tilde{\mathbf{\Lambda}}^{-1} \tilde{\mathbf{y}}) \tag{37}$$

is implemented by solving

$$\mathbf{R}\alpha = \mathbf{D}^T \begin{pmatrix} \mathbf{\Lambda}^{-1/2} \tilde{\mathbf{y}} \\ 0 \end{pmatrix} \tag{38}$$

for $\alpha$ where

$$\mathbf{QR} = \begin{pmatrix} \mathbf{\Lambda}^{-1/2} \hat{\tilde{\mathbf{K}}}_{\mathbf{xz}} \\ \mathbf{L} \end{pmatrix} \tag{39}$$

is the QR decomposition and $\mathbf{K}_{\mathbf{zz}} = \mathbf{L}\mathbf{L}^T$ is the Cholesky decomposition since $\hat{\tilde{\mathbf{C}}} = (\mathbf{QR})^T (\mathbf{QR})$. This has time complexity $\mathcal{O}(nm^2)$ to compute.

### A.3 Implementation

We implement DSoftKI in PyTorch (version 2.4.1) and GPyTorch (Gardner et al., 2018) (version 1.12) so that it can leverage GPU acceleration. DSoftKI currently only leverages a single GPU. Arithmetic for DSoftKI is implemented in single-precision floats. There are a few places where the Cholesky decomposition of $\mathbf{K}_{\mathbf{zz}}$ is utilized in posterior inference. To stabilize these computations, we add a small jitter to the diagonal, when needed. In extreme cases when $\mathbf{K}_{\mathbf{zz}}$ values are extremely small, we may switch to double-precision floats to perform a Cholesky decomposition, before converting the results back into single precision.

Recall that we use CG to implement the Hutchinson pseudoloss. We use a PyTorch implementation of CG descent method following (Maddox et al., 2022). We use a CG tolerance of $1e-5$. We use a rank-10 pivoted Cholseky decomposition to obtain a preconditioner for CG descent.

## B  Additional Experimental Results

We detail the hyperparameters used (Section B.1). We also provide supplemental ablations (Section B.2 and Section B.3) and supplemental experimental results for the experiments described in the main text (Section B.4 and Section B.5).

### B.1 Hyperparameters

For the experiments in this paper, we initialize hyperparameters as follows:

|  | d | Test RMSE | | Test NLL | |
|---|---|---|---|---|---|
|  |  | Exact MLL | Hutchinson | Exact MLL | Hutchinson |
| `Branin` | 2 | **0.002 ± nan** | 0.029 ± 0.0 | **-4.527 ± nan** | 0.208 ± 0.003 |
| `Six-hump-camel` | 2 | **0.014 ± 0.002** | 0.198 ± 0.018 | **-2.185 ± 0.245** | 0.627 ± 0.014 |
| `Styblinski-tang` | 2 | **0.013 ± 0.004** | 0.035 ± 0.001 | **-2.283 ± 0.065** | 0.726 ± 0.08 |
| `Hartmann` | 6 | **0.011 ± 0.003** | 0.408 ± 0.016 | **-3.145 ± 0.318** | 1.159 ± 0.007 |
| `Welch` | 20 | - | **0.618 ± 0.006** | - | **1.373 ± 0.002** |

Table 7: Comparison of using Exact MLL vs. Hutchinson's pseudoloss. The best test RMSE and test NLL are bolded. The notation ±`nan` indicates that one run was unstable whereas the notation − indicates that all runs failed.

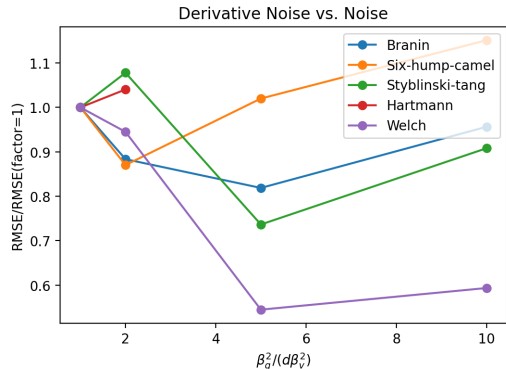

Figure 5: Effect of noise.

- lengthscales $\ell_j = 1$,
- output scale $\gamma = 1$,
- interpolation points $\mathbf{z}$ via $k$-means clustering on training data,
- temperatures $T_{kj} = 1$ for all entries,
- value noise $\beta_v^2 = 0.1$, and
- gradient noise $\beta_g^2 = 0.1 \times d$ where $d$ is the dimensionality of the dataset.

As discussed in the main text, all hyperparameters

$$\theta = (\ell, \gamma, \mathbf{z}^{(m \times d)}, \mathbf{T}^{(m \times d)}, \beta_v, \beta_g) \tag{40}$$

are learned jointly via gradient-based optimization of the stabilized DSoftKI MLL (Equation 25) using Adam (Kingma & Ba, 2014).

### B.2 Exact MLL vs. Hutchinson's Pseudoloss.

Table 7 reproduces the results of Section 5.1 using the Exact MLL compared to Hutchinson's pseudoloss. We see that when using DSoftKI's MLL, there is instability in training on `Branin` and `Welch`. However, using Hutchionson's pseudoloss exclusively leads to worse results. As a result, we opt for an objective that uses the DSoftKI MLL, when possible, and Hutchinson's pseudoloss otherwise as described in Section 4.2.

### B.3 Effect of Noise Levels

The DSoftKI method balances the fitting of value information with gradient information by controlling the relative ratio of $\beta_v^2$ to $\beta_g^2$. Figure 5 shows the results of varying this ratio on the synthetic benchmark to gain more insight into how to set this ratio. We see that the relatative ratio of $\beta_v^2$ to $\beta_g^2$ is dependent on the dataset. The experiments reported in the paper had a relative factor of 1.

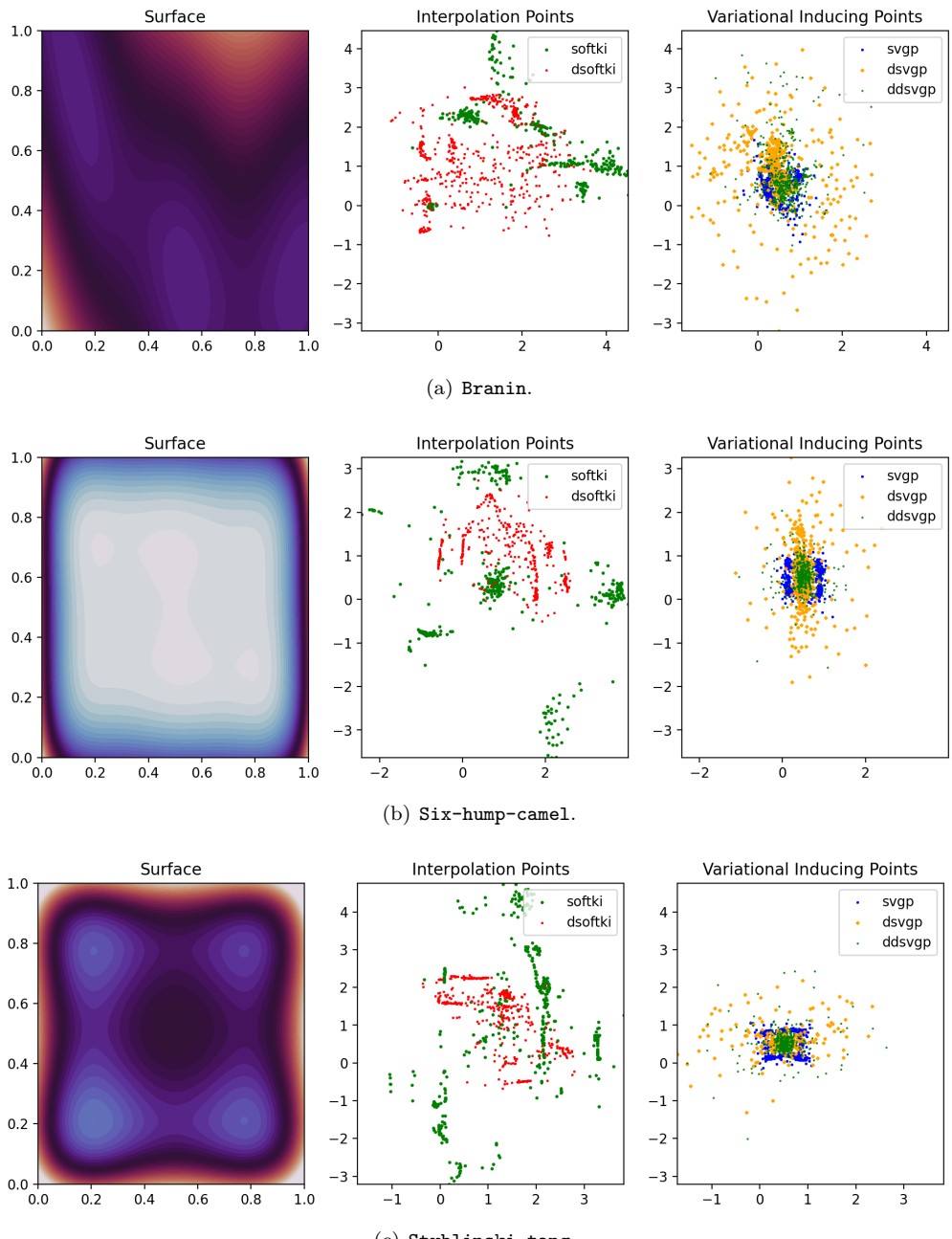

(a) `Branin`.

(b) `Six-hump-camel`.

(c) `Styblinski-tang`.

Figure 6: Learned interpolation and inducing points across various GP and GPwD regression methods.

### B.4 More Synthetic Benchmark Results

**Data normalization.** We scale data $\{(x_i, y_i, dy_i)\}_{i=1}^N$ as $\{(\texttt{hypercube}(x_i), (y_i - \hat{\mu})/\hat{\sigma}, dy_i/\hat{\sigma})\}_{i=1}^N$ where $\texttt{hypercube} : \mathbb{R}^d \rightarrow [0, 1]^d$ maps into the unit hypercube, $\hat{\mu}$ is the empirical mean of the values, and $\hat{\sigma}$ is the empirical standard deviation of the values and the derivatives. This ensures that the values and derivatives are measured in the same units, and follows standard practice.

**Gradient fitting results.** Table 8 reports the gradient fitting errors for the experiment described in Section 5.1 on synthetic functions using scaled effective learning rate. We observe that DSoftKI has the

|  | d | SoftKI | SVGP | DSVGP | DDSVGP | DSoftKI |
|---|---|---|---|---|---|---|
| `Branin` | 2 | $0.17 \pm 0.024$ | $0.237 \pm 0.029$ | $0.357 \pm 0.207$ | $1.526 \pm 0.058$ | $\mathbf{0.07 \pm 0.01}$ |
| `Six-hump-camel` | 2 | $1.069 \pm 0.156$ | $1.206 \pm 0.057$ | $1.545 \pm 0.071$ | $7.032 \pm 0.187$ | $\mathbf{0.345 \pm 0.084}$ |
| `Styblinski-tang` | 2 | $1.008 \pm 0.11$ | $0.998 \pm 0.02$ | $0.86 \pm 0.133$ | $0.895 \pm 0.165$ | $\mathbf{0.319 \pm 0.057}$ |
| `Hartmann` | 6 | $0.429 \pm 0.232$ | $0.377 \pm 0.007$ | $0.506 \pm 0.01$ | $0.527 \pm 0.01$ | $\mathbf{0.028 \pm 0.008}$ |
| `Welch` | 20 | $0.011 \pm$ `nan` | $0.052 \pm 0.0$ | - | $0.269 \pm 0.002$ | $\mathbf{0.001 \pm 0.001}$ |

Table 8: Gradient fitting errors on synthetic functions using scaled effective learning rate. One of the runs for SoftKI encountered numerical instability (hence $\pm$`nan`). $-$ indicates a timeout for DSVGP.

|  | DSVGP | DSoftKI | DSoftKI ($\beta_g^2/(d\beta_v^2) = 10d$) |
|---|---|---|---|
| `Branin` | $\mathbf{0.323 \pm 0.821}$ | $281.505 \pm 72.881$ | $0.397 \pm 0.061$ |
| `Six-hump-camel` | $\mathbf{0.33 \pm 0.248}$ | $53.827 \pm 49.856$ | $8.921 \pm 7.412$ |
| `Styblinski-tang` | $\mathbf{0.665 \pm 0.121}$ | $41.907 \pm 14.305$ | $4.929 \pm 2.126$ |
| `Hartmann` | $\mathbf{0.047 \pm 0.014}$ | $10.755 \pm 0.916$ | $0.591 \pm 0.265$ |
| `Welch` | - | $-1.329 \pm 1.906$ | $\mathbf{0.138 \pm 0.161}$ |

Table 9: Uncertainty quantification over gradient predictions.

lowest gradient errors of the methods. The relative size of $\beta_v^2$ and $\beta_g^2$ controls the degree to which value fitting is prioritized compared to gradient fitting.

We also compare the gradient test RMSE of the base GP and its extension with derivatives. For a gradient test RMSE, we sum up the errors across the dimensions. Thus, we should normalize by the number of dimensions to compare across datasets. We observe that it is dataset dependent whether or not DSVGP outperforms SVGP. As a reminder, DDSVGP does not fit all the derivative information, and thus, we derive its gradient prediction from its value prediction. We observe that derivative information is helpful for DSoftKI, the extension of SoftKI to the setting with derivatives.

**Uncertainty quantification of derivatives.** We only report test NLLs for DSVGP and DSoftKI since these are the only methods that provide uncertainty estimates for full derivative information. In general, we find that DSoftKI has larger test NLLs compared to DSVGP. This is consistent with our earlier findings that DSVGP gives lower test NLL compared to SVGP, even though its test RMSE is worse.

**Visualization.** Figure 6 visualizes the learned interpolation/inducing points for each method. We make a couple of observations on the learned points. First, we see that there are differences between the structure of the interpolation points learned by SoftKI and DSoftKI compared to the inducing points learned by the variational approaches. In particular, the variational approaches have additional learned parameters that can help represent the structure in the dataset. In contrast, SoftKI and DSoftKI have to rely solely on the interpolation points to represent the structure in the dataset. Second, we observe that the interpolation/inducing points learned those GPs that fit derivatives and those that do not are different. In particular, DSoftKI and SoftKI capture different geometric structure in the underlying data. The inducing points learned by DSVGP are more spread out compared to those learned by DDSVGP and SVGP.

## B.5 More Force Field Results

**Dataset details.** More concretely, we have a labeled data set $\{(x_i, y_i, dy_i)\}_i$ where $x_i$ encodes the *Cartesian coordinates* of the atomic nuclei in the molecule in Ångström's, $y_i$ encodes the energy of the molecule in kcal/mol, and $dy_i$ encodes the negative force exerted on each atom in the molecule. The dimensionality of $x_i$ is $d = 3A$ where $A$ is the number of atoms in the molecule since each atom takes 3 coordinates to describe in 3*D*-space. For example, a 42 atom system will have $d = 126$. We consider all available molecules.

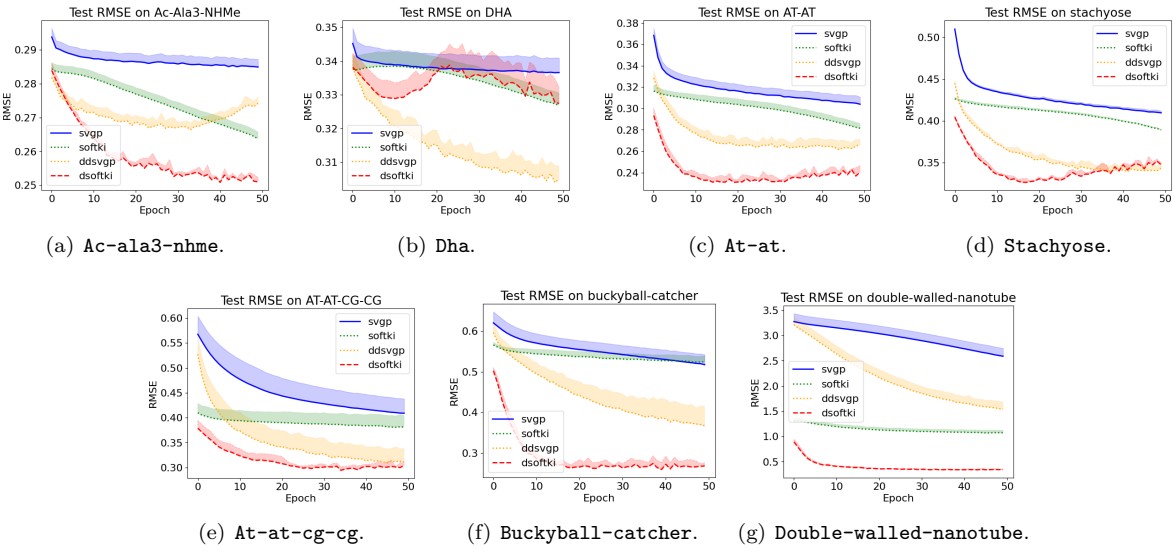

Figure 7: Test RMSE curves.

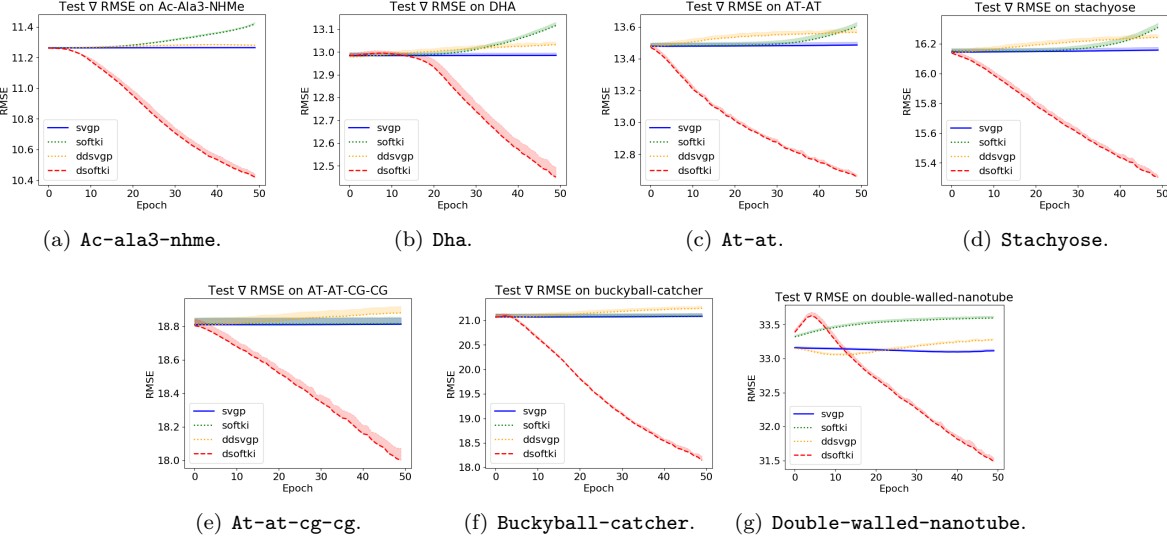

Figure 8: Test gradient RMSE curves.

**Data normalization.** We use the same data normalization scheme for energies and forces as we did in the synthetic experiments. Instead of using `hypercube`, we scale the Cartesian coordinates as $x_i/3$ which has the effect of changing units.

**Training curves.** Figure 7 and Figure 8 give the test RMSE, test gradient RMSE, and their standard deviations at each epoch of training on the MD22 dataset. As a reminder, we did not tune the learning rates or batch sizes to obtain the best possible performance. Rather, we control the effective learning rate so that we can compare GP and GPwD regression. SVGP is in blue, SoftKI is in green, DDSVGP is in orange, and DSoftKI is in Red. On `Ac-Ala3-NHMe`, we observe that methods that fit derivative information overfit both values and gradients. On several datasets such as `AT-AT` and `Stachyose`, we observe that DSoftKI overfits on test RMSE but does not on test gradient RMSE. On other dataset such as `AT-AT-CG-CG`, `Buckyball-catcher`, and `Double-walled-nanotube`, we observe that DSoftKI is able to fit both the values and gradients. In contrast, DDSVGP tends to overfit as measured by test gradient RMSE. This is somewhat

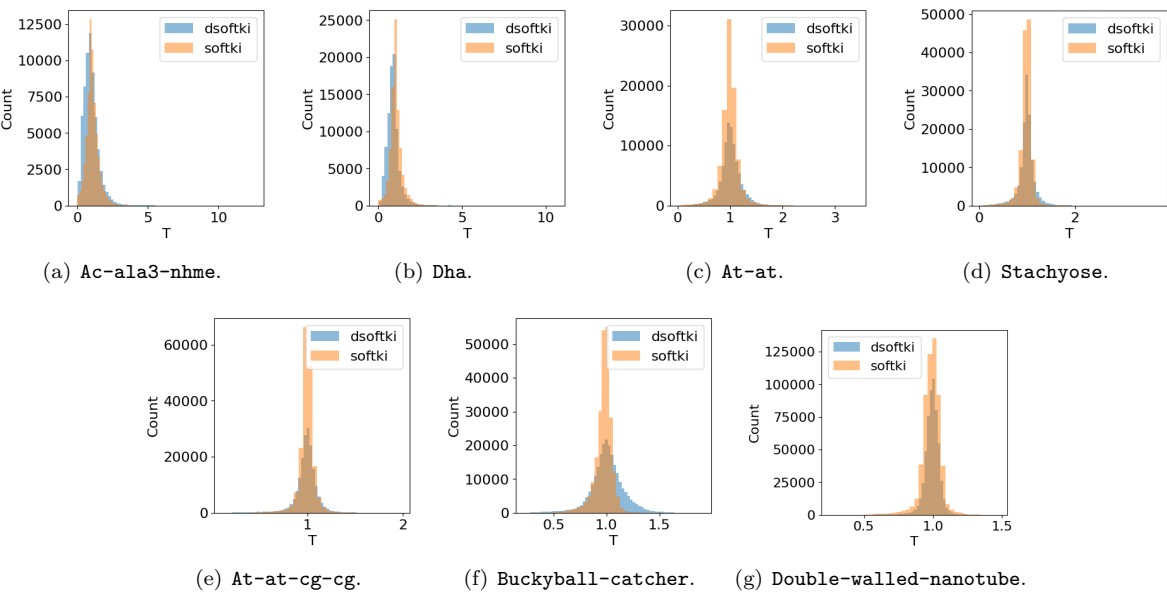

Figure 9: Histogram of learned temperatures for DSoftKI and SoftKI on the MD22 dataset. The learned temperatures have similar distributions.

surprising, since it does fit directional derivative information. We also observe that SoftKI and SVGP cannot fit gradients even though they do fit the values. This suggests that GPs that do not fit derivatives do not capture the shape of the surface well.

**Temperatures and lengthscales.** Figure 9 and Figure 10 give the histograms of temperatures and lengthscales that are learned by DSoftKI and SoftKI. As a reminder, we modified SoftKI to also have different temperatures per interpolation point as we did for DSoftKI to have a more fair comparison.

The temperature $T_k \in \mathbb{R}^d$ for interpolation point $z_k$ controls how strongly that point responds to variations along each input dimension during interpolation. A smaller temperature component $T_{kj}$ makes the interpolation weight $\sigma_z^j(x)$ more sensitive to changes in dimension $j$, effectively sharpening the directional influence. Conversely, larger temperatures smooth out the response along that dimension.

Figure 9 shows that the learned temperature distributions are similar between DSoftKI and SoftKI, indicating that the interpolation scheme (which is what is affected by the temperature) is primarily adapted to the data geometry. Moreover, we observe that DSoftKI learns larger kernel lengthscales compared to SoftKI (Figure 10). This suggests that when gradient information is available, the model can rely on the temperature vectors to capture local directional variation, allowing the kernel lengthscales to increase and model smoother global structure. In other words, the temperatures and lengthscales play complementary roles: temperatures handle local directional sensitivity while lengthscales control global smoothness.

## C   Additional Datasets

We describe experiments on additional datasets including a toy n-body simulation (Section C.1) and the UCI dataset with synthetic gradients (Section C.2) to test DSoftKI on a broader range of datasets.

### C.1   Toy N-Body Dataset

Computational physics and chemistry provide settings where high-dimensional gradient observations arise naturally. Consider an $n$-body system with $n$ particles (*e.g.*, atoms or point masses). Let $\mathbf{q} = (q_1, \ldots, q_n)$ denote the positions of each particle where each $q_i \in \mathbb{R}^3$, and let $\mathbf{p} = (p_1, \ldots, p_n)$ denote the momenta where

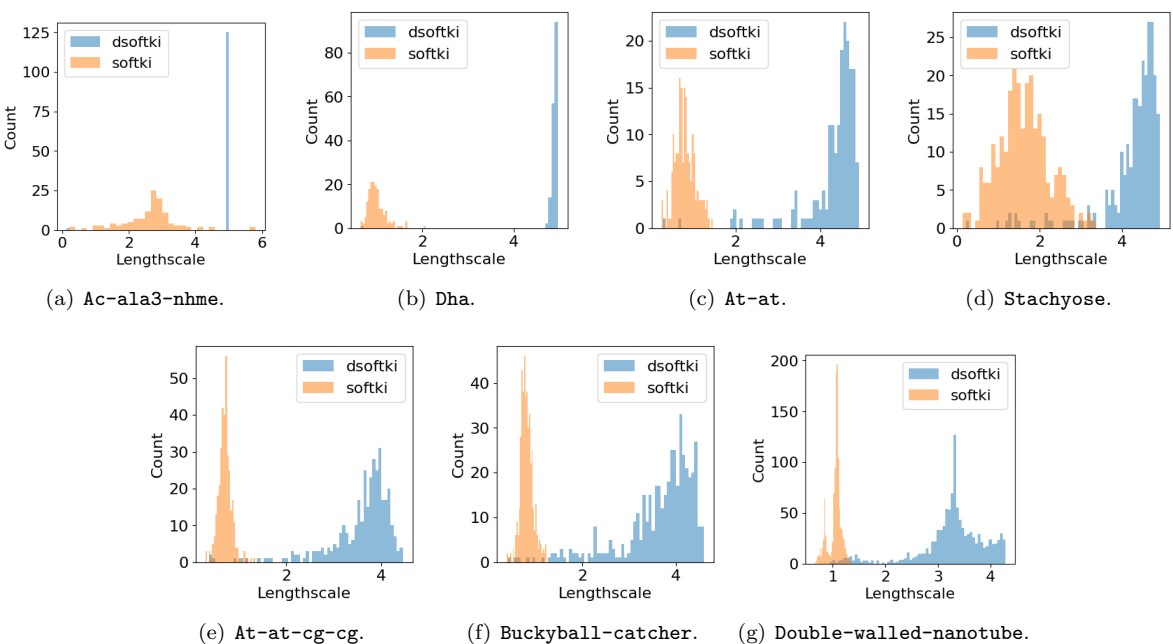

Figure 10: Histogram of lengthscales on MD22 dataset for DSoftKI and SoftKI. The lengthscales learned by the DSoftKI tend to be larger than the lengthscales learned by SoftKI.

| | d | DDSVGP | DSoftKI | $\nabla$-DDSVGP | $\nabla$-DSoftKI |
|---|---|---|---|---|---|
| nbody-4 | 24 | **0.058 ± 0.005** | 0.059 ± 0.002 | 1.453 ± 0.083 | **0.708 ± 0.034** |
| nbody-6 | 36 | **0.027 ± 0.001** | 0.035 ± 0.002 | 1.5 ± 0.059 | **1.032 ± 0.045** |
| nbody-8 | 48 | 0.03 ± 0.0 | **0.028 ± 0.002** | 1.368 ± 0.117 | **1.016 ± 0.092** |
| nbody-10 | 60 | 0.026 ± 0.003 | **0.024 ± 0.0** | 1.343 ± 0.037 | **1.104 ± 0.04** |

Table 10: Test RMSE (best bolded) on $n$-body datasets. $\nabla$-DDSVGP and $\nabla$-DSoftKI denote gradient RMSE (best bolded).

each $p_i \in \mathbb{R}^3$. The dynamics are governed by Hamilton's equations

$$\frac{d\mathbf{q}}{dt} = \frac{\partial H}{\partial \mathbf{p}}, \quad \frac{d\mathbf{p}}{dt} = -\frac{\partial H}{\partial \mathbf{q}} \tag{41}$$

where $H(\mathbf{q}, \mathbf{p})$ is the *Hamiltonian*. Thus, given observations of the Hamiltonian $H$ and its gradients $\nabla H$, we can learn the system's dynamics, *i.e.*, $\frac{d\mathbf{q}}{dt}$ and $\frac{d\mathbf{p}}{dt}$. More concretely, it is a GPwD regression problem where $\mathbf{x} = (\mathbf{q}, \mathbf{p})$, $\mathbf{y} = H(\mathbf{x})$, and

$$\nabla \mathbf{y} = \left( \frac{\partial H}{\partial \mathbf{q}}, \frac{\partial H}{\partial \mathbf{p}} \right) . \tag{42}$$

The input dimensionality is $d = 6n$ since each particle contributes 3 position and 3 momentum coordinates. A learned surrogate for the Hamiltonian enables efficient prediction of the system's dynamics without expensive numerical integration, which can be intractable for large $n$.

We construct an $n$-body gravitational dataset `nbody-n` as follows.[3] The Hamiltonian is

$$H(\mathbf{q}, \mathbf{p}) = \sum_{i=1}^{n} \frac{\|p_i\|^2}{2m_i} - \sum_{i<j} \frac{Gm_im_j}{\sqrt{\|q_i - q_j\|^2 + \epsilon^2}} \tag{43}$$

---

[3]This is a classical system, in contrast to the MD22 dataset (Section 5.2) which is a quantum system.

|  | d | DDSVGP | DSoftKI |
|---|---|---|---|
| nbody-4 | 24 | **0.98 ± 0.016** | 1.68 ± 0.019 |
| nbody-6 | 36 | **0.995 ± 0.029** | 2.802 ± 0.048 |
| nbody-8 | 48 | **0.97 ± 0.006** | 3.646 ± 0.059 |
| nbody-10 | 60 | **0.976 ± 0.001** | 4.591 ± 0.162 |

Table 11: Training time per epoch (in seconds, best bolded) on $n$-body Hamiltonian datasets.

|  | n | d | DDSVGP | DSoftKI | $\nabla$-DDSVGP | $\nabla$-DSoftKI |
|---|---|---|---|---|---|---|
| Kin40k | 18000 | 8 | 0.867 ± 0.002 | **0.864 ± 0.006** | 0.399 ± 0.001 | **0.393 ± 0.001** |
| Protein | 20578 | 9 | 0.81 ± 0.006 | **0.787 ± 0.003** | 2.778 ± 0.087 | **2.655 ± 0.283** |
| Bike | 7820 | 17 | 0.629 ± 0.026 | **0.421 ± 0.006** | 0.763 ± 0.004 | **0.624 ± 0.012** |
| Elevators | 7470 | 18 | 0.821 ± 0.013 | **0.797 ± 0.009** | 0.449 ± 0.002 | **0.448 ± 0.001** |
| Pol | 6750 | 26 | 0.898 ± 0.007 | **0.702 ± 0.01** | 0.383 ± 0.013 | **0.324 ± 0.012** |
| Slice | 19260 | 385 | 0.357 ± 0.005 | **0.03 ± 0.008** | 0.165 ± 0.007 | **0.094 ± 0.004** |

Table 12: Test RMSE (best bolded) on selected UCI datasets. $\nabla$-DDSVGP and $\nabla$-DSoftKI denote gradient RMSE (best bolded).

where $m_i$ is the mass of particle $i$, $G$ is the gravitational constant, and $\epsilon$ is a softening parameter that prevents singularities when particles approach each other. The first term is the kinetic energy and the second is the gravitational potential energy with Plummer softening (Plummer, 1911). In this toy example, the Hamiltonian has analytic gradients which may not exist for more complex systems.

We generate datasets for $n \in \{4, 6, 8, 10\}$ particles, corresponding to input dimensions $d \in \{24, 36, 48, 60\}$. To generate physically realistic configurations, we simulate 100 trajectories by integrating Hamilton's equations using a high-order Runge-Kutta method (DOP853) as implemented in SciPy (Virtanen et al., 2020). Initial positions are sampled from $\mathcal{N}(0, 2^2)$, initial momenta from $\mathcal{N}(0, 0.5^2)$ scaled by particle mass, and masses uniformly from $[0.5, 2.0]$. We apply center-of-mass corrections for numerical stability. From each trajectory, we sample configurations to obtain 10000 total samples per setting, computing the Hamiltonian and its gradients analytically. We filter out the 5% of samples with the largest gradient norms for numerical stability, leaving 9500 samples. For this toy dataset, we use $G = 1$ and $\epsilon = 0.1$.

Table 10 reports the results of using a 90/10 training-testing split on each dataset. DSoftKI and DDSVGP achieve similar value RMSE, but DSoftKI substantially outperforms DDSVGP on gradient prediction (*e.g.*, 0.708 RMSE versus 1.453 $\nabla$-RMSE for $n = 4$). This demonstrates DSoftKI's advantage in applications requiring accurate full gradient predictions, such as learning physical dynamics from energy observations, and justifies the higher computational cost (Table 11).

## C.2 UCI Dataset

To evaluate scaling on diverse input distributions beyond physics datasets, we construct GPwD regression benchmarks from standard UCI regression datasets (Dua & Graff, 2017) by generating synthetic gradients. We select six UCI datasets spanning a range of dimensions: Kin40k ($d = 8$), Protein ($d = 9$), Bike ($d = 17$), Elevators ($d = 18$), Pol ($d = 26$), and Slice ($d = 385$). For each dataset, we estimate gradients using a $k$-nearest neighbor finite difference approximation: for each point $x_i$ with label $y_i$, we compute directional derivatives along the directions to its $k = 3$ nearest neighbors and average the resulting gradient estimates, *i.e.*,

$$\nabla y_i \approx \frac{1}{k} \sum_{j \in N_k(i)} \frac{y_j - y_i}{\|x_j - x_i\|^2}(x_j - x_i) \tag{44}$$

where $N_k(i)$ denotes the $k$ nearest neighbors of point $i$. This introduces noise into the gradient observations, providing a test of robustness to gradient estimation error.

|            | n     | d   | DDSVGP            | DSoftKI            |
|------------|-------|-----|-------------------|--------------------|
| Kin40k     | 18000 | 8   | $2.257 \pm 0.022$ | $\mathbf{1.715 \pm 0.1}$ |
| Protein    | 20578 | 9   | $2.593 \pm 0.05$  | $\mathbf{2.054 \pm 0.067}$ |
| Bike       | 7820  | 17  | $\mathbf{0.977 \pm 0.003}$ | $1.317 \pm 0.067$ |
| Elevators  | 7470  | 18  | $\mathbf{0.994 \pm 0.024}$ | $1.297 \pm 0.017$ |
| Pol        | 6750  | 26  | $\mathbf{0.876 \pm 0.014}$ | $1.663 \pm 0.077$ |
| Slice      | 19260 | 385 | $\mathbf{2.378 \pm 0.047}$ | $71.279 \pm 0.076$ |

Table 13: Wall-clock training time (in seconds) per epoch (best bolded) on selected UCI datasets.

|                        | DSoftKI                          | DSoftKI-DKL                |
|------------------------|----------------------------------|----------------------------|
| Ac-ala3-nhme           | $\mathbf{5.98e\text{-}03 \pm 2.90e\text{-}05}$ | $6.93e\text{-}03 \pm 9.40e\text{-}05$ |
| Dha                    | $\mathbf{5.85e\text{-}03 \pm 8.80e\text{-}05}$ | $6.19e\text{-}03 \pm 7.40e\text{-}05$ |
| At-at                  | $\mathbf{4.01e\text{-}03 \pm 1.04e\text{-}04}$ | $5.63e\text{-}03 \pm 5.40e\text{-}05$ |
| Stachyose              | $\mathbf{3.99e\text{-}03 \pm 2.90e\text{-}05}$ | $5.29e\text{-}03 \pm 1.30e\text{-}05$ |
| At-at-cg-cg            | $\mathbf{2.58e\text{-}03 \pm 5.60e\text{-}05}$ | $3.98e\text{-}03 \pm 2.14e\text{-}04$ |
| Buckyball-catcher      | $\mathbf{1.81e\text{-}03 \pm 5.40e\text{-}05}$ | $4.09e\text{-}03 \pm 1.17e\text{-}04$ |
| Double-walled-nanotube | $\mathbf{9.33e\text{-}04 \pm 2.70e\text{-}05}$ | $2.90e\text{-}03 \pm 4.59e\text{-}04$ |

Table 14: Normalized test RMSE per atom (best bolded) on MD22 dataset.

Table 12 reports the results. DSoftKI outperforms DDSVGP on both value and gradient prediction across all datasets. The Slice dataset ($d = 385$) is particularly notable: DSoftKI achieves 0.030 RMSE versus DDSVGP's 0.357, demonstrating strong performance on high-dimensional structured data. The higher training time for DSoftKI on Slice (Table 13) reflects its $O(m^2nd)$ complexity, but is justified by substantially better accuracy.

## D  Deep Kernel Learning

One advantage of DSoftKI over DSVGP and DDSVGP is that it does not require computing first or second-order derivatives of the kernel (Section 4.1). This enables the use of learned kernels, such as Deep Kernel Learning (Wilson et al., 2016) (DKL), without hard-coding kernel derivatives so that standard automatic differentiation techniques can be applied.

To demonstrate this capability, we evaluate DSoftKI with a deep kernel learning (DSoftKI-DKL) setup on the MD22 molecular datasets. We use a 2-layer MLP feature extractor with hidden dimension 64, output dimension 24, and tanh activations. The RBF kernel is applied in the learned feature space. We train all hyperparameters jointly using Adam, with a learning rate of 0.002 for the GP hyperparameters and 0.0005 for the neural network weights.

Table 14, Table 15, and Table 16 report the results. DSoftKI-DKL performs comparably to DSoftKI, with slightly higher RMSE on both values and gradients. We emphasize that this experiment is intended as a demonstration that deep kernel learning works out of the box with DSoftKI, rather than a claim that it improves performance on these particular datasets. Notably, DSoftKI-DKL offers computational speedups over DSoftKI since the interpolation points are projected into a lower-dimensional feature space. Furthermore, the neural network gradients can be computed efficiently via Jacobian-vector products, whereas DSVGP and DDSVGP would require hard-coding the feature extractor Jacobians or face intractable computation.

## E  Additional Background on DDSVGP and DSKI

In this section, we describe in more detail how variational inducing point methods and SKI have been extended to the setting with derivatives.

|                        | DSoftKI                      | DSoftKI-DKL          |
|------------------------|------------------------------|----------------------|
| `Ac-ala3-nhme`         | **8.27e-02 $\pm$ 1.73e-04**  | 8.62e-02 $\pm$ 1.00e-05 |
| `Dha`                  | **7.41e-02 $\pm$ 2.59e-04**  | 7.62e-02 $\pm$ 2.27e-04 |
| `At-at`                | **7.03e-02 $\pm$ 8.50e-05**  | 7.49e-02 $\pm$ 1.25e-04 |
| `Stachyose`            | **5.86e-02 $\pm$ 6.00e-05**  | 6.19e-02 $\pm$ 1.04e-04 |
| `At-at-cg-cg`          | **5.08e-02 $\pm$ 2.02e-04**  | 5.32e-02 $\pm$ 1.10e-04 |
| `Buckyball-catcher`    | **4.09e-02 $\pm$ 1.78e-04**  | 4.75e-02 $\pm$ 1.66e-04 |
| `Double-walled-nanotube` | **2.84e-02 $\pm$ 5.10e-05** | 3.28e-02 $\pm$ 1.04e-03 |

Table 15: Normalized gradient test RMSE per dimension $d$ (best bolded) on MD22 dataset.

|                        | n     | d    | DSoftKI            | DSoftKI-DKL          |
|------------------------|-------|------|--------------------|----------------------|
| `Ac-ala3-nhme`         | 76598 | 126  | 82.665 $\pm$ 0.06  | **58.98 $\pm$ 0.516** |
| `Dha`                  | 62777 | 168  | 89.155 $\pm$ 0.223 | **48.736 $\pm$ 0.56** |
| `At-at`                | 18000 | 180  | 27.236 $\pm$ 0.037 | **13.958 $\pm$ 0.025** |
| `Stachyose`            | 24544 | 261  | 54.181 $\pm$ 0.142 | **19.324 $\pm$ 0.369** |
| `At-at-cg-cg`          | 9137  | 354  | 27.102 $\pm$ 0.036 | **12.574 $\pm$ 0.339** |
| `Buckyball-catcher`    | 5491  | 444  | 20.764 $\pm$ 0.111 | **7.655 $\pm$ 0.146** |
| `Double-walled-nanotube` | 4528 | 1110 | 45.379 $\pm$ 0.007 | **7.143 $\pm$ 0.088** |

Table 16: Comparing (wall-clock) training time (in seconds) per epoch (best bolded) on MD22 dataset for DSoftKI versus DSoftKI-DKL.

**Variational inducing points.** An *inducing point method* (Snelson & Ghahramani, 2005; Quinonero-Candela & Rasmussen, 2005) introduces a set of $m \ll n$ *inducing points* $\mathbf{z} = (z_i \in \mathbb{R}^d)_{i=1}^m$ and associated *inducing variables* $f(\mathbf{z}) = \mathbf{u} = (u_i \in \mathbb{R})_{i=1}^m$ as a proxy for the given inputs $\mathbf{x}$ and outputs $\mathbf{y}$ respectively. In an inducing point method such as Sparse Variational Gaussian Processes (SGPR) (Titsias, 2009) and Stochastic Variational Gaussian Process (SVGPs) (Hensman et al., 2013), the inducing points and variables are related to the dataset as

$$\begin{pmatrix} \mathbf{u} \\ f(\mathbf{x}) \end{pmatrix} \sim \mathcal{N}\left(0, \begin{pmatrix} \mathbf{K_{zz}} & \mathbf{K_{zx}} \\ \mathbf{K_{xz}} & \mathbf{K_{xx}} \end{pmatrix}\right) \qquad \text{(inducing)}$$

$$\mathbf{y} \mid f(\mathbf{x}) \sim \mathcal{N}(f(\mathbf{x}), \mathbf{\Lambda}). \qquad \text{(likelihood)}$$

Note that the marginalization of the inducing variables $\mathbf{u}$ reduces the model to the standard GP model. To make posterior inference tractable, a SVGP makes a variational approximation $q(f(\mathbf{x}), \mathbf{u}) = p(f(\mathbf{x}) \mid \mathbf{u})q(\mathbf{u})$ where $q(\mathbf{u}) = \mathcal{N}(\mathbf{u} \mid \mathbf{m}, \mathbf{S})$, treating $\mathbf{m}^{(m \times 1)}$ and $\mathbf{S}^{(m \times m)}$ as additional learnable variational parameters. More concretely, $\theta = (\ell, \gamma, \beta, \mathbf{z}, \mathbf{m}, \mathbf{S})$. They can be learned by maximizing a lower bound on the MLL called the evidence lower bound (ELBO), defined as

$$\text{ELBO}(q(f(\mathbf{x}), \mathbf{u})) = \sum_{i=1}^n \mathbb{E}_{q(f(x_i))} p(y_i \mid f(x_i)) - KL(q(\mathbf{u}) \,\|\, p(\mathbf{u} \mid \mathbf{z})) \qquad (45)$$

which equivalently minimizes the KL-divergence $KL(q(\mathbf{u}) \,\|\, p(\mathbf{u} \mid \mathbf{z}))$. The ELBO can be optimized in mini-batches, resulting in a time complexity of $O(m^3)$. Given learned hyperparameters, the optimal posterior distribution is

$$q(f(*)) = \mathcal{N}(f(*) \mid \mathbf{K_{*z}}\mathbf{K_{zz}^{-1}}\mathbf{m}, \mathbf{K_{**}} - \mathbf{K_{*z}}\mathbf{K_{zz}^{-1}}(\mathbf{S} - \mathbf{K_{zz}})\mathbf{K_{zz}^{-1}}\mathbf{K_{z*}} + \mathbf{\Lambda}). \qquad (46)$$

The complexity of inference is $O(m^3)$.

A SVGP can be extended to the setting with derivatives, *i.e.*, a DSVGP (Padidar et al., 2021). More concretely, define an inducing variable $\tilde{u}_i = \begin{pmatrix} u_i & \nabla u_i^T \end{pmatrix}^T$ that models the value and its gradient at an

inducing point $z_i$. A DSVGP introduces learnable parameters $\mathbf{m}^{(m \times 1)}$, $d\mathbf{m}^{(dm \times 1)}$, $\mathbf{S}^{(m \times m)}$, $d\mathbf{S}^{(1 \times m)}$, and $d^2\mathbf{S}^{(dm \times dm)}$ to define a variational posterior $q(\tilde{\mathbf{u}}) = \mathcal{N}(\tilde{\mathbf{u}} \,|\, \tilde{\mathbf{m}}, \tilde{\mathbf{S}})$ where

$$\tilde{\mathbf{m}} = \left[ \begin{pmatrix} \mathbf{m}_i \\ d\mathbf{m}_i \end{pmatrix} \right]_{i1} \text{ and } \tilde{\mathbf{S}} = \left[ \begin{pmatrix} \mathbf{S}_{ij} & d\mathbf{S}_{ij} \\ d\mathbf{S}_{ij} & d^2\mathbf{S}_{ij} \end{pmatrix} \right]_{ij}. \tag{47}$$

The parameters can be learned by maximizing the DSVGP ELBO which is obtained by replacing the SVGP ELBO with the respective $\tilde{(\cdot)}$ versions.

Again, it is a lower-bound on the MLL. The time complexity of computing the ELBO on a minibatch is $O(m^3 d^3)$. The posterior distribution is that of SVGP with all of the matrices replaced with their respective $\tilde{(\cdot)}$ versions. The time complexity of posterior inference is $O(m^3 d^3)$ since $\tilde{K}_{\mathbf{zz}}$ is a $(m(d+1)) \times (m(d+1))$ matrix.

DDSVGP (Padidar et al., 2021) utilizes directional derivatives $\partial_{\bar{\mathbf{V}}} f = \bar{\mathbf{V}}^T \nabla f(x)$ where $\bar{\mathbf{V}}$ projects onto a $p$ dimensional subspace. This results in the modified matrices

$$\bar{\mathbf{K}}_{\mathbf{zz}} = \left[ \begin{pmatrix} \mathbb{I} \\ \partial_{\mathbf{V}_a} \end{pmatrix} k(z_a, z_b) \begin{pmatrix} \mathbb{I} & \partial_{\mathbf{V}_b}^T \end{pmatrix} \right]_{ab} \tag{48}$$

$$\bar{\mathbf{K}}_{*\mathbf{z}} = \left[ \begin{pmatrix} \mathbb{I} \\ \nabla_* \end{pmatrix} k(*, z_b) \begin{pmatrix} \mathbb{I} & \partial_{\mathbf{V}_b}^T \end{pmatrix} \right]_{1b} \tag{49}$$

which has additional learnable variational parameters $\{\bar{\mathbf{V}}_a^{(d \times p)}\}_{a=1}^m$ for projecting. The $m(p+1) \times m(p+1)$ matrix $\bar{\mathbf{K}}_{\mathbf{zz}} = [\bar{k}(z_a, z_b)]_{ab}$ can be formulated efficiently with Hessian-vector products in time complexity $\mathcal{O}(m^2 dp)$. The modified posterior distribution and ELBO are obtained by changing the respective $(\cdot)$ variables to the $(\bar{\cdot})$ versions. Posterior inference can be computed in time $\mathcal{O}(m^3 p^3)$, assuming $mp^2 > d$ to account for the cost of forming $\bar{\mathbf{K}}_{\mathbf{zz}}$, since the $d$-dimensional gradients have been reduced to $p$-dimensional directional derivatives.

**Structured kernel interpolation.** One problem with inducing point methods is the requirement that $m \ll n$. SKI (Wilson & Nickisch, 2015) overcomes this limitation by using the approximate kernel

$$\mathbf{K}_{\mathbf{xx}}^{\text{SKI}} = \mathbf{W}_{\mathbf{xz}} \mathbf{K}_{\mathbf{zz}} \mathbf{W}_{\mathbf{zx}} \approx \mathbf{K}_{\mathbf{xx}} \tag{50}$$

where $\mathbf{z}$ are cleverly chosen interpolation points on a pre-defined lattice, $\mathbf{W}_{\mathbf{xz}} = [w_{\mathbf{z}}^j(x_i)]_{ij}$ is a matrix of interpolation weights with interpolation function $w_{\mathbf{z}}^j(x_i)$, and $\mathbf{W}_{\mathbf{xz}} = \mathbf{W}_{\mathbf{zx}}^T$. The SKI posterior is

$$p(f(*) \,|\, \mathbf{x}, \mathbf{y}) = \mathcal{N}(f(*) \,|\, \mathbf{K}_{*\mathbf{x}}(\mathbf{K}_{\mathbf{xx}}^{\text{SKI}} + \mathbf{\Lambda})^{-1}\mathbf{y}, \mathbf{K}_{**} - \mathbf{K}_{*\mathbf{x}}(\mathbf{K}_{\mathbf{xx}}^{\text{SKI}} + \mathbf{\Lambda})^{-1}\mathbf{K}_{\mathbf{x}*}) \tag{51}$$

which is the GP posterior with $\mathbf{K}_{\mathbf{xx}}$ replaced with $\mathbf{K}_{\mathbf{xx}}^{\text{SKI}}$.

The posterior is tractable to compute since the lattice structure given by the interpolation points enables fast matrix-vector multiplies (MVMs) so that conjugate gradient (CG) methods can be used to solve large systems of linear equations. The complexity of a single MVM is $O(n4^d + m \log m)$ with a cubic interpolation function (Wilson & Nickisch, 2015) where $m$ is the number of interpolation points. However, the asymptotic dependence on $d$ limits the application of SKI to low dimensions.

DSKI (Eriksson et al., 2018) extends the SKI (Wilson & Nickisch, 2015) method to create a scalable GP regression method that handles derivative information. To accomplish this, DSKI uses the following approximate kernel

$$\tilde{\mathbf{K}}_{\mathbf{xx}}^{\text{DSKI}} = \tilde{\mathbf{W}}_{\mathbf{xz}} \mathbf{K}_{\mathbf{zz}} \tilde{\mathbf{W}}_{\mathbf{zx}} \approx \tilde{\mathbf{K}}_{\mathbf{xx}} \tag{52}$$

where

$$\tilde{\mathbf{W}}_{\mathbf{xz}} = \left[ \begin{pmatrix} w_{\mathbf{z}}^j(x_i) \\ \nabla w_{\mathbf{z}}^j(x_i) \end{pmatrix} \right]_{ij} = \left[ \begin{pmatrix} \mathbb{I} \\ \nabla_{x_i} \end{pmatrix} w_{\mathbf{z}}^j(x_i) \right]_{ij}. \tag{53}$$

That is, DSKI approximates the kernel by differentiating the interpolation matrix. The posterior distribution is the SKI posterior with the SKI kernel replaced with the DSKI kernel. Like SKI, DSKI leverages fast MVMs enabled by the lattice structure and CG methods to perform GP inference. The complexity of a single MVM is $O(nd6^d + m \log m)$. The scaling in the dimension $d$ is even worse than SKI, limiting the application of DSKI to even smaller dimensions.

