# OpenReview forum: "Scaling Gaussian Process Regression with Full Derivative Observations"
_TMLR — Accepted by TMLR_

### Review · Reviewer_sBME · 2025-10-04

**Summary Of Contributions:**

This paper focuses on Gaussian Process Regression in a particular setting - when the derivatives of the target function are observed (GPwD), and proposes a novel technique called DSoftKI to achieve it more efficiently than existing techniques that is particularly relevant for high-dimensional data. This technique is summarized in Algo 1, and argued to be more efficient than SoftKI, an existing technique to achieve the same. The heart of Gaussian Process lies in the Kernel, and hence the core approach here is "Soft Kernel Interpolation with Derivatives" with the help of interpolation points, that also introduces "temperature" variables. In a regression setting, the variables are estimated in a Gradient-descent based algorithm, where the gradients of the objective function (p(y|x,M) is estimated using Hutchinson pseudoloss. In the proposed approach, this is calculated efficiently, using a Cholesky Decomposition and a low-rank reconstruction of the Kernel matrix over the training points. The posterior inference formula is calculated accordingly. Experimental analysis compare the RMSE and Negative Log-Likelihood achieved by the proposed method vis-a-vis benchmarks on benchmark synthetic datasets, and also on high-dimensional molecular force fields.

In general, the paper and the contributions are sound, though the main weaknesses in my opinion are as follows:

The technical contribution may be on the lighter side, as it seems to be a (non-trivial) increment over SoftKI
The experimental validation is mainly on simulated datasets, except one molecular force field data

**Audience:**

Yes

**Audience Explanation:**

Gaussian Process regression is an important approach in many domain, and there are many open computational questions related to it that interests the ML community

**Claims And Evidence:**

No

**Claims Explanation:**

My main concern is that most of the experiments are on synthetic datasets except one molecular force field data.  Also, there are insufficient experimental results related to computational efficiency, which is the main motivation of the paper

**Requested Changes:**

My main suggestion would be to consider a couple of more datasets from different scientific domains where Gradient information of the target function is known, and show the results. Furthermore, the main claim of the proposed algorithm is computational efficiency, so there should be more detailed comparisons of running time for the proposed approach against the benchmarks.

One feature of the proposed approach is the additional temperature variables. The appendix does show some histograms of these variables, but we would like to understand their physical significance - which interpolation points are assigned higher/lower temperatures, and how these affect the final result.

---

> ### Author Response · Authors · 2025-12-19
> **Response**
>
> > My main suggestion would be to consider a couple of more datasets from different scientific domains where Gradient information of the target function is known, and show the results.
>
> Thank you for this suggestion. We agree that additional real-world datasets would strengthen the empirical validation.
> We note that datasets with true and high-dimensional gradient observations are relatively rare outside of computational chemistry/physics, where gradients arise naturally as forces from energy calculations, and are synthetically constructed. We introduce a n body gravitational Hamiltonian in 3 dimensions where x = (position, momentum), y = Hamiltonian(position, momentum), and dy/dx = (d_Hamiltonian/d_position, d_Hamiltonian/d_momentum).
>
> On the N-body Hamiltonian dataset, DSoftKI and DDSVGP achieve similar value RMSE, but DSoftKI substantially outperforms DDSVGP on gradient prediction (e.g., 0.708 vs 1.453 ∇RMSE for Nbody-4). This demonstrates DSoftKI's advantage in applications requiring accurate full gradient predictions.
>
> |          |    N |   D | DSoftKI           | DDSVGP            | DRMSE-DSoftKI         | DRMSE-DDSVGP          | time-DSoftKI      | time-DDSVGP       |
> |:---------|-----:|----:|:------------------|:------------------|:------------------|:------------------|:------------------|:------------------|
> | Nbody-4  | 9000 |  24 | 0.059 $\pm$ 0.002 | 0.058 $\pm$ 0.005 | 0.708 $\pm$ 0.034 | 1.453 $\pm$ 0.083 | 1.68 $\pm$ 0.019  | 0.98 $\pm$ 0.016  |
> | Nbody-6  | 9000 |  36 | 0.035 $\pm$ 0.002 | 0.027 $\pm$ 0.001 | 1.032 $\pm$ 0.045 | 1.5 $\pm$ 0.059   | 2.802 $\pm$ 0.048 | 0.995 $\pm$ 0.029 |
> | Nbody-8  | 9000 |  48 | 0.028 $\pm$ 0.002 | 0.03 $\pm$ 0.0    | 1.016 $\pm$ 0.092 | 1.368 $\pm$ 0.117 | 3.646 $\pm$ 0.059 | 0.97 $\pm$ 0.006  |
> | Nbody-10 | 9000 |  60 | 0.024 $\pm$ 0.0   | 0.026 $\pm$ 0.003 | 1.104 $\pm$ 0.04  | 1.343 $\pm$ 0.037 | 4.591 $\pm$ 0.162 | 0.976 $\pm$ 0.001 |
>
> Additionally, to evaluate scaling on diverse input distributions beyond physics datasets, we have also included UCI regression benchmarks with synthetically generated gradients via finite differences. This setup allows assessment of computational scaling on other datasets with noisy gradients to test DSoftKI’s performance on this area.
>
> |          |    N |   D | DSoftKI           | DDSVGP            | DRMSE-DSoftKI         | DRMSE-DDSVGP          | time-DSoftKI      | time-DDSVGP       |
> |:---------|-----:|----:|:------------------|:------------------|:------------------|:------------------|:------------------|:------------------|
> | Kin40k    | 18000 |   8 | 0.864 $\pm$ 0.006 | 0.867 $\pm$ 0.002 | 0.393 $\pm$ 0.001 | 0.399 $\pm$ 0.001 | 1.715 $\pm$ 0.1    | 2.257 $\pm$ 0.022 |
> | Protein   | 20578 |   9 | 0.787 $\pm$ 0.003 | 0.81 $\pm$ 0.006  | 2.655 $\pm$ 0.283 | 2.778 $\pm$ 0.087 | 2.054 $\pm$ 0.067  | 2.593 $\pm$ 0.05  |
> | Bike      |  7820 |  17 | 0.421 $\pm$ 0.006 | 0.629 $\pm$ 0.026 | 0.624 $\pm$ 0.012 | 0.763 $\pm$ 0.004 | 1.317 $\pm$ 0.067  | 0.977 $\pm$ 0.003 |
> | Elevators |  7470 |  18 | 0.797 $\pm$ 0.009 | 0.821 $\pm$ 0.013 | 0.448 $\pm$ 0.001 | 0.449 $\pm$ 0.002 | 1.297 $\pm$ 0.017  | 0.994 $\pm$ 0.024 |
> | Pol       |  6750 |  26 | 0.702 $\pm$ 0.01  | 0.898 $\pm$ 0.007 | 0.324 $\pm$ 0.012 | 0.383 $\pm$ 0.013 | 1.663 $\pm$ 0.077  | 0.876 $\pm$ 0.014 |
> | Slice     | 19260 | 385 | 0.03 $\pm$ 0.008  | 0.357 $\pm$ 0.005 | 0.094 $\pm$ 0.004 | 0.165 $\pm$ 0.007 | 71.279 $\pm$ 0.076 | 2.378 $\pm$ 0.047 |
>
> On UCI benchmarks with synthetic gradients, DSoftKI outperforms DDSVGP on gradient prediction (DRMSE) cross all datasets. The Slice dataset (D=385) is particularly notable: DSoftKI achieves 0.030 RMSE versus DDSVGP's 0.357, demonstrating strong performance on high-dimensional structured data. The higher training time for DSoftKI on Slice reflects its $O(mnd^2)$ complexity, but is justified by substantially better accuracy. We will add this to the appendix.

---

> ### Author Response · Authors · 2025-12-19
> **Response**
>
> > Furthermore, the main claim of the proposed algorithm is computational efficiency, so there should be more detailed comparisons of running time for the proposed approach against the benchmarks.
>
> Thank you for this question. We will put the following tables in the appendix due to space limitations in the main text.
>
> Table A: Training time per epoch (seconds) on synthetic benchmarks
> |                 |     N |   D | SoftKI      | SVGP         | DSVGP        | DDSVGP       | DSoftKI      |
> |:----------------|------:|----:|:------------------|:------------------|:------------------|:------------------|:------------------|
> | Branin          | 10000 |   2 | 0.194 $\pm$ 0.005 | 0.339 $\pm$ 0.149 | 1.318 $\pm$ 0.02  | 1.945 $\pm$ 0.022 | 2.555 $\pm$ 0.199 |
> | Six-hump-camel  | 10000 |   2 | 0.257 $\pm$ 0.092 | 0.373 $\pm$ 0.104 | 1.31 $\pm$ 0.008  | 1.955 $\pm$ 0.062 | 2.434 $\pm$ 0.087 |
> | Styblinski-tang | 10000 |   2 | 0.21 $\pm$ 0.006  | 0.335 $\pm$ 0.14  | 1.316 $\pm$ 0.018 | 1.91 $\pm$ 0.016  | 2.539 $\pm$ 0.137 |
> | Hartmann        | 10000 |   6 | 0.199 $\pm$ 0.003 | 0.337 $\pm$ 0.126 | 9.305 $\pm$ 0.113 | 1.934 $\pm$ 0.045 | 2.544 $\pm$ 0.091 |
> | Welch           | 10000 |  20 | 0.213 $\pm$ 0.002 | 0.418 $\pm$ 0.123 | -                 | 1.944 $\pm$ 0.063 | 4.452 $\pm$ 0.192 |
>
> Table B: Training time per epoch (seconds) on MD22 molecular datasets
> |                        |     N |    D | time-soft-gp      | time-svgp         | time-ddsvgp       | time-dsoftki       |
> |:-----------------------|------:|-----:|:------------------|:------------------|:------------------|:-------------------|
> | Ac-ala3-nhme           | 76598 |  126 | 0.894 $\pm$ 0.006 | 1.051 $\pm$ 0.018 | 7.907 $\pm$ 0.004 | 82.665 $\pm$ 0.06  |
> | Dha                    | 62777 |  168 | 0.887 $\pm$ 0.021 | 0.951 $\pm$ 0.027 | 6.544 $\pm$ 0.012 | 89.155 $\pm$ 0.223 |
> | At-at                  | 18000 |  180 | 0.336 $\pm$ 0.006 | 0.427 $\pm$ 0.001 | 2.626 $\pm$ 0.01  | 27.236 $\pm$ 0.037 |
> | Stachyose              | 24544 |  261 | 0.475 $\pm$ 0.005 | 0.555 $\pm$ 0.007 | 3.344 $\pm$ 0.021 | 54.181 $\pm$ 0.142 |
> | At-at-cg-cg            |  9137 |  354 | 0.355 $\pm$ 0.008 | 0.423 $\pm$ 0.007 | 1.988 $\pm$ 0.006 | 27.102 $\pm$ 0.036 |
> | Buckyball-catcher      |  5491 |  444 | 0.244 $\pm$ 0.007 | 0.295 $\pm$ 0.004 | 1.249 $\pm$ 0.007 | 20.764 $\pm$ 0.111 |
> | Double-walled-nanotube |  4528 | 1110 | 0.409 $\pm$ 0.007 | 0.434 $\pm$ 0.004 | 1.906 $\pm$ 0.003 | 45.379 $\pm$ 0.007 |
>
> DSoftKI has higher per-epoch training time than DDSVGP, particularly for large d. This is expected given DSoftKI's $O(m^2nd)$ complexity versus DDSVGP's $O(m^3p^3)$ with $p=2$. However, two key points justify this cost:
> 1. Full gradient prediction: DSoftKI is the only method that predicts all d gradient components. DDSVGP only predicts p=2 directional derivatives, which is insufficient for applications like molecular dynamics where full forces are required since the shape of the surface also needs to be fit.
> 2. DSVGP intractability: DSVGP, which does predict full gradients, becomes intractable for $d>20$ (note the missing entries and the jump to 9.3 seconds at $d=6$). DSoftKI remains tractable up to $d=1110$.
>
> In summary, DSoftKI trades per-iteration speed for the ability to predict full gradients at scales where no other method can operate.

---

> ### Author Response · Authors · 2025-12-19
> **Response**
>
> > One feature of the proposed approach is the additional temperature variables. The appendix does show some histograms of these variables, but we would like to understand their physical significance - which interpolation points are assigned higher/lower temperatures, and how these affect the final result.
>
> Thank you for this question. The temperature vectors serve to encode directional sensitivity for each interpolation point, allowing nearby interpolation points to capture different gradient directions. The temperature $T_k \in \mathbb{R}^d$ for interpolation point $z_k$ controls how strongly that point responds to variations along each input dimension during interpolation. A smaller temperature component $T_{kj}$ makes the interpolation weight $\sigma_z^j(x)$ more sensitive to changes in dimension $j$ effectively sharpening the directional influence. Conversely, larger temperatures smooth out the response along that dimension.
>
> Interestingly, Figure 10 shows that the learned temperature distributions are similar between DSoftKI and SoftKI, indicating that the interpolation scheme (which is what is affected by the temperature) is primarily adapted to the data geometry. Additionally, from Figure 9 in the appendix, we observe that DSoftKI learns larger kernel lengthscales compared to SoftKI. This suggests that when gradient information is available, the model can rely on the temperature vectors to capture local directional variation, allowing the kernel lengthscales to increase and model smoother global structure. In other words, the temperatures and lengthscales play complementary roles: temperatures handle local directional sensitivity while lengthscales control global smoothness. We will expand upon this discussion in the appendix and also in the main text.

---

> > ### Comment · Reviewer_sBME · 2026-01-02
> >
> > I thank the authors for their response. The paper definitely looks much stronger with these additional results.

---

> > > ### Author Response · Authors · 2026-01-06
> > >
> > > We thank the reviewer again for their time and thoughtful questions/comments.

---

### Review · Reviewer_csfB · 2025-12-07

**Summary Of Contributions:**

This paper introduces a GP capable of fitting and predicting full derivative information, called DSoftKI, by extending SoftKI. This very recent method approximates a kernel via softmax interpolation from learned interpolation point locations. The method achieves a posterior inference time complexity of $O(ndm^2)$, allowing it to scale to larger values of $n$ and $d$ than previous GPwD approaches while still retaining the ability to model full derivative information. Here $m$ is the number of inducing points (Quinonero-Candela & Rasmussen, 2005; Snelson & Ghahramani, 2005). The authors evaluate DSoftKI on GPwD regression tasks and demonstrate that it achieves strong accuracy, as measured by both test RMSE and test NLL, while effectively modeling gradients.

In general, the idea of the paper is not novel. The time complexity of $O(ndm^2)$ directly comes from the original approach, SoftKI.  However, the approach that DSoftKI handles derivative observations is non-trivial, and it entails several substantive methodological contributions.

**Audience:**

Yes

**Audience Explanation:**

The topic of Scaling Gaussian Process Regression using derivative observations is important. Although the proposed DSoftKI method builds upon SoftKI, extending it to handle derivative observations is non-trivial and entails several methodological contributions.

**Claims And Evidence:**

Yes

**Claims Explanation:**

- To implement the softmax interpolation scheme, they associate each interpolation point with a corresponding learnable
temperature vector as in Equation 20.
- DSoftKI introduces a different interpolation scheme and more learnable parameter at Algorithm 1.
- Section 4.3 introduces the choice of value and gradient noises.

**Requested Changes:**

- How to estimate the variable $T$?
- It is unclear why DSoftKI is more efficient than DDSVGP? As the author explained in the introduction, introducing $p <<d$ from DDSVGP comes at the cost of directly predicting derivatives and introducing further approximations. However, their method also adds many approximations.

---

> ### Author Response · Authors · 2025-12-19
> **Response**
>
> We thank the reviewer for this question. The temperature matrix $\mathbf{T}\in R^{(m×d)}$, where each interpolation point $z_k$​ has an associated temperature vector $\mathbf{T}_k \in \mathbb{R}^d$, is learned via gradient-based optimization of the stabilized DSoftKI marginal log-likelihood (Equation 22), jointly with the other hyperparameters $\theta = (\ell, \gamma, \mathbf{z}, \mathbf{T}, \beta_v^2, \beta_g^2)$. This is described in Section 4.2 (Hyperparameter Optimization) and summarized in Algorithm 1, Line 7. We will clarify this more explicitly in the revised manuscript by adding a sentence stating that all components of $\theta$, including $\mathbf{T}$, are optimized jointly using Adam.
>
> > It is unclear why DSoftKI is more efficient than DDSVGP? As the author explained in the introduction, introducing $p \ll d$ from DDSVGP comes at the cost of directly predicting derivatives and introducing further approximations. However, their method also adds many approximations.
>
> Thank you for this important question. While both DSoftKI and DDSVGP introduce approximations, they differ fundamentally in what is approximated and the consequences for scalability and prediction.
> Approximation strategy:
> - *DSVGP/DDSVGP*: Introduces separate inducing points for values and gradients, resulting in a kernel matrix of size $m(d+1)×m(d+1)m(d+1) \times m(d+1)$ for DSVGP. To avoid cubic scaling in dd d, DDSVGP further approximates the gradient space by projecting onto $p \ll d$  inducing directions, resulting in a kernel matrix of size $m(p+1)×m(p+1)$.
> - *DSoftKI*: Uses the same interpolation points to jointly model values and gradients through an interpolated kernel. The gradient information is incorporated via the interpolation weights $\tilde{\Sigma}_{xz}$ (Equation 15) of size $m \times n(d+1)$ and maintains the a kernel matrix $K_{zz}$ of size $m \times m$, not through separate inducing variables.
>
> Consequences:
> - Gradient prediction*: DSoftKI predicts full d-dimensional gradients. DDSVGP only predicts p-dimensional directional derivatives, losing fidelity (as illustrated in Figure 1, where DDSVGP fails to capture the gradient structure).
> Kernel derivatives: DSVGP/DDSVGP require computing first and second-order derivatives of the kernel, which must be hard-coded for each kernel type. DSoftKI approximates these via interpolation, enabling use with arbitrary kernels including learned kernels.
> - *Complexity*: DSoftKI achieves $O(m^2nd)$ posterior inference without introducing additional inducing directions, whereas DDSVGP's reported $O(m^3p^3)$ complexity assumes $mp^2 > d$ (see footnote 1).
>
> We will clarify these distinctions more explicitly in the revised manuscript. We will also emphasize in the main text that appendix C contains more details on comparing the DSoftKI and DSVGP/DDSVGP kernels.

---

> > ### Comment · Reviewer_csfB · 2026-01-05
> >
> > I thank the reviewers for addressing my concerns. The contribution of the paper is now much clearer.

---

> > > ### Author Response · Authors · 2026-01-06
> > >
> > > We thank the reviewer again for their time and thoughtful questions/comments.

---

### Review · Reviewer_Dm1z · 2025-12-10

**Summary Of Contributions:**

Key contribution: the paper presents a new scalable GP method for fitting functions when noisy measurements of both function evaluations and the corresponding gradients are available. The method is a generalization of SoftKI (Camaño & Huang, 2025). As its predecessor, the method does not require computing kernel first and second order gradients, which improves computational complexity; unlike SoftKI, the interpolation-point specific temperature vectors allow more flexibility in modeling the gradients, which requires changes in the implementational considerations that keep the computations scalable. Small dimensional  and high dimensional data is used to show the benefits brought about by the introduction of the temperature hyperparameters.

**Audience:**

Yes

**Audience Explanation:**

The results are of broad relevance for GP applications with high dimensional data and gradient information (e.g. in the natural sciences); of particular utility might be scenarios when kernels need to be flexible (Wilson et al., 2016).

**Broader Impact Concerns:**

No concerns

**Claims And Evidence:**

Yes

**Claims Explanation:**

The methodology and its relations to previous methods is rigorously presented in the main text

The text is also thorough in justifying the exact setup for the numerical comparisons, the complexities entailed in the comparison and the limitations of the approach.

**Requested Changes:**

I found the text generally very readable and the descriptions precise and rigurous.

A few suggestions of improvement:

1) in section 3.3 softmax: i found the /T notation is weird for a d-dimensional vector T
2) would be useful to remind people of the dimensionality of variables in section 4

Practical notes:
- any specific considerations about the initial values of the hyperparameters?
- in empirical data i expect the gradient measurements to be generally noisier  that the primary observations, how does that affect the estimation
- since the text emphasizes that the method is one of the few that can be used with learned kernels it would be useful to include a concrete demonstration of that
- was not sure where the heteroskedastic noise was for, please explain

---

> ### Author Response · Authors · 2025-12-19
> **Response**
>
> > A few suggestions of improvement: in section 3.3 softmax: i found the /T notation is weird for a d-dimensional vector T
>
> We thank the reviewer for this suggestion. We agree the notation $x/T$ is unclear for vector division. We will switch to the notation
>
> $x \odiv T$ where $\odiv$ is a Hadamard division (i.e., element-wise division).
>
> > any specific considerations about the initial values of the hyperparameters?
>
> We thank the reviewer for this question. We initialize hyperparameters as follows:
> Interpolation points z: k-means clustering on training data
> Lengthscales ℓ: 1.0
> Output scale γ: 1.0
> Temperatures T: 1.0 for all entries
> Value noise βv: 0.1
> Gradient noise βg: 0.1 × d (following the ratio discussion in Section 4.3)
> We found these initializations to be robust across our experiments. More sophisticated initialization strategies, such as placing priors on hyperparameters or using empirical Bayes, could be explored in future work.
>
> > in empirical data i expect the grsdient measurements to be generally noisier that the primary observations, how does that affect the estimation
>
> Thank you for this important question. DSoftKI handles settings where gradient observations are noisier through separate noise parameters $\beta_v^2$ and $\beta_g^2$. As discussed in Section 4.3, the ratio $\beta_g^2/\beta_v^2$ controls the relative weighting of value versus gradient fitting. When gradients are noisier (larger $\beta_g$), the model appropriately downweights their contribution. In our experiments, we found that setting $\beta_g^2/\beta_v^2 \approx d$ (dimension of dataset) worked well as a baseline, though practitioners should adjust this ratio based on their domain knowledge of relative measurement reliability. Figure 5 in the appendix shows how varying this ratio affects DSoftKI's performance, demonstrating that prioritizing value fitting generally improves results on the synthetic benchmarks. We have also run experiments with noisy and finite difference gradients on the UCI dataset (see response to Reviewer sBME)
>
> > since the text emphasizes that the method is one of the few taht can be used with learned kernels it would be useful to include a concrete demonstration of that
>
> Thank you for this suggestion. We created a simple deep kernel learning (DKL) experiment where DSoftKI-DKL uses a learned feature extractor (2-layer MLP) with a hidden dimension of 64, output dimension of 24, and tanh activations. We train all hyperparameters jointly, and use a learning rate of 0.002 for the GP hyperparameters and and 0.0005 for learning the embedding. DSoftKI-DKL performs slightly worse than DSoftKI in terms of accuracy. We emphasize that this setup is intended as a demonstration of kernel learning “working out of the box with DSoftKI.”
>
> |                        |   RMSE-DSoftKI |   RMSE-DSoftKI-DKL |   DRMSE-DSoftKI |   DRMSE-DSoftKI-DKL |
> |:-----------------------|----------:|--------------:|----------------:|--------------------:|
> | Ac-ala3-nhme           |    0.0020 |        0.0023 |          0.0827 |              0.0862 |
> | Dha                    |    0.0019 |        0.0021 |          0.0741 |              0.0762 |
> | At-at                  |    0.0013 |        0.0019 |          0.0703 |              0.0749 |
> | Stachyose              |    0.0013 |        0.0018 |          0.0586 |              0.0619 |
> | At-at-cg-cg            |    0.0009 |        0.0013 |          0.0508 |              0.0532 |
> | Buckyball-catcher      |    0.0006 |        0.0014 |          0.0409 |              0.0475 |
> | Double-walled-nanotube |    0.0003 |        0.0010 |          0.0284 |              0.0328 |
>
> We also report timing. We note that DSoftKI-DKL presents speedups over DSoftKI since the interpolation points are now projected into a lower-dimensional space. Moreover, the neural network weights can be obtained by Jacobian vector products. This is a key advantage compared to DSVGP/DDSVGP which would be intractable to compute or require hard-coding feature extraction Jacobians.
>
> |                        | time-DSoftKI       | time-DSoftKI-DKL   |
> |:-----------------------|:-------------------|:-------------------|
> | Ac-ala3-nhme           | 82.665 $\pm$ 0.06  | 58.98 $\pm$ 0.516  |
> | Dha                    | 89.155 $\pm$ 0.223 | 48.736 $\pm$ 0.56  |
> | At-at                  | 27.236 $\pm$ 0.037 | 13.958 $\pm$ 0.025 |
> | Stachyose              | 54.181 $\pm$ 0.142 | 19.324 $\pm$ 0.369 |
> | At-at-cg-cg            | 27.102 $\pm$ 0.036 | 12.574 $\pm$ 0.339 |
> | Buckyball-catcher      | 20.764 $\pm$ 0.111 | 7.655 $\pm$ 0.146  |
> | Double-walled-nanotube | 45.379 $\pm$ 0.007 | 7.143 $\pm$ 0.088  |

---

> ### Author Response · Authors · 2025-12-19
> **Response**
>
> > was not sure where the heteroskedastic noise was for, please explain
>
> Thank you for this question. We used the term "heteroskedastic" when describing the PLL modification for the variational baselines (DSVGP, DDSVGP), following the terminology in Padidar et al. (2021). In this context, it refers to allowing separate noise parameters for function values ($\beta_v^2$) versus gradient observations ($\beta_g^2$​), rather than a single shared noise parameter. DSoftKI similarly uses separate noise parameters for values and gradients (Section 4.3). We do not model noise that varies across different input locations. We will clarify this terminology in the revised manuscript to avoid confusion, since we agree that this is not heteroskedastic in the traditional sense.

---

### Author Response · Authors · 2025-12-19
**Response**

We thank all reviewers for their constructive feedback. We address the key concerns below.

Concern on novelty:
We respectfully clarify that while DSoftKI builds upon SoftKI, the extension to derivative observations is non-trivial and introduces several methodological contributions:
1. Per-interpolation point temperature vectors: The key technical innovation is associating each interpolation point with a learnable temperature vector $T_k \in \mathbb{R}^d$ that encodes directional sensitivity. This allows nearby interpolation points to capture different gradient directions—a capability absent in SoftKI. As shown in Table 4, using the original SoftKI interpolation scheme (shared temperature) performs substantially worse, demonstrating that this modification is essential for modeling derivatives.
2. First scalable kernel interpolation method for GPwD with full gradient prediction: Prior kernel interpolation methods for derivatives (DSKI) scale as $O(nd^{6d})$ limiting them to very low dimensions. Variational methods either have cubic scaling in d (DSVGP) or sacrifice full gradient prediction (DDSVGP with $p \ll d$ directions). DSoftKI achieves $O(m^2nd)$. while predicting all d gradient components—a unique capability at high dimensions (up to $d=1110$ in our experiments).
3. No kernel derivatives required: Unlike DSVGP/DDSVGP, DSoftKI approximates kernel derivatives via interpolation rather than computing them analytically. This enables use with arbitrary kernels, including learned kernels, without hard-coding derivatives. We will add experiments to the appendix to show the deep kernel learning approach (see response for Reviewer Dm1z).
4. Joint noise regularization requires new analysis: The unified posterior coefficients (Equation 25) are jointly influenced by value and gradient noise, requiring new analysis of the relative regularization $\beta_g^2/\beta_v^2$ ratio (Section 4.3) not present in SoftKI. This structure also differs from DSVGP/DDSVGP which use separate inducing variables for values and gradients., as opposed to DSoftKI which uses the same interpolation values for values and gradients.

We also note that extending methods to handle derivatives is a recognized contribution—DSVGP/DDSVGP extend SVGP, DSKI extends SKI.

Concern on computational efficiency comparisons (see response for Reviewer sBME):
We acknowledge that runtime comparisons were missing from the original submission. We will add timing tables in the appendix showing per-epoch training time across all methods on both MD22 and synthetic benchmarks (see below). The results show that while DSoftKI has higher per-iteration cost than DDSVGP, it is the only method capable of predicting full gradients at high dimensions. DSVGP becomes intractable for $d>20$, while DSoftKI scales to $d = 1110$.

---

### Comment · Reviewer_Dm1z · 2026-01-02
**Manuscript substantially improved**

I thank the authors for the thorough revision. I think this is a very solid publication and have no additional questions or concerns.

---

> ### Author Response · Authors · 2026-01-06
>
> We thank the reviewer again for their time and thoughtful questions/comments.

---

### Decision · Action_Editor_jCeZ · 2026-01-11

**Recommendation:** Accept with minor revision

**Additional Comments:**

All reviewers' concerns have been well addressed.  Please update the paper to include the results shown in the discussion with reviewers.

**Audience:**

Yes

**Audience Explanation:**

Scalable GPwD is important tool at least in quantum chemistry.  All reviewers acknowledged the significance of the paper.

**Claims And Evidence:**

Yes

**Claims Explanation:**

The proposed method and its relation to previous methods are clearly explained and the performance was demonstrated as claimed in abstract and introduction.  All reviewers agreed that the claims were well supported.